# A Graph Meta-Network for Learning on Kolmogorov-Arnold Networks

**Guy Bar-Shalom**[*]     **Ami Tavory**     **Itay Evron**     **Maya Bechler-Speicher**
Technion, Meta          Meta              Meta              Meta

**Ido Guy**                                    **Haggai Maron**
Meta, Ben-Gurion University of the Negev         Technion

## Abstract

Weight-space models learn directly from the parameters of neural networks, enabling tasks such as predicting their accuracy on new datasets. Naive methods – like applying MLPs to flattened parameters – perform poorly, making the design of better weight-space architectures a central challenge. While prior work leveraged permutation symmetries in standard networks to guide such designs, no analogous analysis or tailored architecture yet exists for Kolmogorov–Arnold Networks (KANs). In this work, we show that KANs share the same permutation symmetries as MLPs, and propose the *KAN-graph*, a graph representation of their computation. Building on this, we develop WS-KAN, the first weight-space architecture that learns on KANs, which naturally accounts for their symmetry. We analyze WS-KAN's expressive power, showing it can replicate an input KAN's forward pass - a standard approach for assessing expressiveness in weight-space architectures. We construct a comprehensive "zoo" of trained KANs spanning diverse tasks, which we use as benchmarks to empirically evaluate WS-KAN. Across all tasks, WS-KAN consistently outperforms structure-agnostic baselines, often by a substantial margin. Our code is available at https://github.com/BarSGuy/KAN-Graph-Metanetwork.

## 1 Introduction

Deep neural networks are now powerful tools for prediction, generation, and beyond. A recent perspective (Navon et al., 2023; Zhang et al., 2023) views their parameters not merely as weights, but as data – enabling the design of *weight-space models*, which are networks that operate directly on the parameters of other networks. This shift makes it possible to predict test accuracy (Eilertsen et al., 2020; Unterthiner et al., 2020), generate new sets of weights (Erkoç et al., 2023), and classify or synthesize Implicit Neural Representations (INRs; Mescheder et al. 2019; Sitzmann et al. 2020), all through a single forward pass.

A straightforward way to design weight-space (WS) models is to flatten all weights and biases into a single feature vector for prediction. However, this overlooks *neuron permutation symmetries* (Hecht-Nielsen, 1990; Brea et al., 2019), i.e., parameter transformations that keep the underlying function computed by the neural network unchanged. Thus, such naive models may produce different predictions for equivalent reorderings. While pioneering approaches used generic architectures to process model parameters (Schürholt et al., 2022c;b;e), more recent developments incorporate those networks' symmetries, either through weight sharing in linear layers (Navon et al., 2023; Zhou et al., 2023) (via geometric deep learning principles; Bronstein et al. 2021), or by treating networks as graphs (Lim et al., 2024; Kalogeropoulos et al., 2024; Zhang et al., 2023; Kofinas et al., 2024) and applying Graph Neural Networks (GNNs; Scarselli et al. 2008; Kipf 2016; Gilmer et al. 2017), which naturally respect these symmetries.

Previous work on WS learning has only begun to explore the potential of characterizing and applying WS models across different architectures. Initially, research has concentrated on MLPs with

---

[*]This work was completed during an internship at Meta.

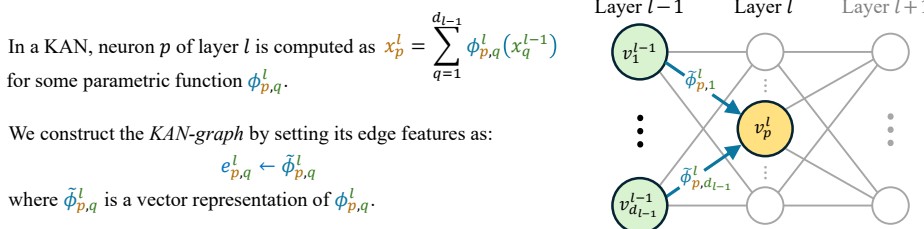

In a KAN, neuron $p$ of layer $l$ is computed as $x_p^l = \sum_{q=1}^{d_{l-1}} \phi_{p,q}^l(x_q^{l-1})$ for some parametric function $\phi_{p,q}^l$.

We construct the *KAN-graph* by setting its edge features as:

$$e_{p,q}^l \leftarrow \tilde{\phi}_{p,q}^l$$

where $\tilde{\phi}_{p,q}^l$ is a vector representation of $\phi_{p,q}^l$.

Figure 1: Constructing the KAN-graph for a given Kolmogorov-Arnold Network (KAN).

straightforward extensions to CNNs (Navon et al., 2023; Zhou et al., 2023). Recent developments have expanded to transformer architectures (Tran et al., 2024) and architectures incorporating tensor symmetry structures (Zhou et al., 2024). However, designing WS models for diverse architectural paradigms remains an important open challenge that warrants further investigation.

In this work, we introduce the first WS model specifically designed to process an emerging class of neural networks: Kolmogorov–Arnold Networks (KANs; Liu et al. 2025). *Why design a weight-space model that takes KANs as input?* KANs represent a fundamentally different neural paradigm—rather than employing scalar weight matrices with fixed nonlinear activations, they construct networks from matrices of learnable univariate functions, while preserving the universal approximation properties of conventional neural networks (Kolmogorov, 1954; Braun & Griebel, 2009). This architectural shift yields compelling advantages over standard MLPs, including superior parameter efficiency (Koenig et al., 2025), accelerated neural scaling (i.e., performance scales faster than expected as model size increases) (Liu et al., 2025), and notably, enhanced interpretability (Barašin et al., 2024). The learnable functions that replace scalar weights can be directly visualized and analyzed, offering unprecedented insight into the network's decision-making process.

As KANs gain traction within the deep learning community, we anticipate a proliferation of trained KAN models across diverse applications. WS learning will become increasingly valuable for helping practitioners understand, compare, and leverage these models effectively. However, developing weight-space models for KANs presents several challenges. The network architecture differs fundamentally from those previously studied in the weight-space literature, as its learnable components are functions rather than simple scalar parameters. Additionally, the structural understanding from symmetry perspectives that have been developed for traditional neural networks remains largely unexplored for KANs.

**Our contributions.** We begin by showing that KANs also exhibit permutation symmetries – in fact, the same as conventional neural networks. Building on this insight, we introduce the *KAN-graph*, a novel attributed graph with edge features that compactly encodes the structure of a given KAN; see Figure 1. On top of this representation, we develop WS-KAN, a GNN-based architecture capable of learning directly over the *KAN-graph*. We show that WS-KAN applied to a KAN-graph can simulate the forward pass of the corresponding KAN, validating our approach from a theoretical perspective, and laying the ground for stronger results such as functional approximation theorems. To validate our approach experimentally, we construct the first "model zoo" of pre-trained KANs across diverse tasks, together with their corresponding KAN-graph representations serving as a benchmark. We show that WS-KAN consistently outperforms both generic baselines (such as MLPs over flattened parameters) and more sophisticated architectures, which effectively act as ablation studies and further validate our architectural design from the perspective of the symmetry analysis. Furthermore, because WS-KAN operates directly on graph representations through a GNN-based architecture, it can be seamlessly applied to KAN architectures of varying sizes, including those not encountered during training. We empirically demonstrate promising results in these settings.

## 2 BACKGROUND AND RELATED WORK

**Equivariance and invariance in deep learning.** Many learning tasks involve functions that either remain unchanged (invariant) or transform in a predictable manner (equivariant) under specific symmetries of the input. A classic example is the translation invariance of Convolutional Neural Networks (CNN; Krizhevsky et al. 2012), where shifting an image does not alter its label. While

simple MLPs can, in principle, learn such symmetries from data, this approach is often inefficient. By explicitly incorporating the symmetry into the model architecture Cohen et al. (2018); Maron et al. (2020); Zaheer et al. (2017); Ravanbakhsh et al. (2017), the property becomes an inherent feature of the model rather than something that must be inferred during training. This typically leads to better generalization and data efficiency (Cohen & Welling, 2016; Brehmer et al., 2024). Building on these principles, a new class of models has recently emerged, referred to as *weight-space models*.

**weight-space models.** The *weight-space* of a neural network refers to the collection of parameters that fully define its architecture. For a multilayer perceptron (MLP), this is the set of weights and biases across all layers, $\theta = (\mathbf{W}_1, \mathbf{b}_1, \ldots, \mathbf{W}_L, \mathbf{b}_L)$. *weight-space models* are approaches that operate directly on this parameter space. While pioneering approaches proposed standard architectures for learning in WS (Schürholt et al., 2021; 2022d;a), more recent works (Navon et al., 2023; Zhou et al., 2023) have focused on the symmetries inherent in neural networks (Hecht-Nielsen, 1990; Brea et al., 2019), leading to tailored architectures that explicitly respect these symmetries. Several strategies have emerged: for instance, Navon et al. (2023); Zhou et al. (2023) enforce weight sharing in linear layers, while *Graph Meta-Networks* (Kalogeropoulos et al., 2024; Kofinas et al., 2024; Lim et al., 2024) leverage graph neural networks to operate directly on a model's computational graph. In particular, Lim et al. (2024) has shown that those neural network symmetries correspond to graph automorphisms of their computational graphs.

**Kolmogorov–Arnold Networks (KANs).** KANs are a recently introduced class of neural networks in which edges of the computational graph, rather than carrying simple numerical weights, abstractly represent univariate functions (Liu et al., 2025). Formally, given an input $\boldsymbol{x} \in \mathbb{R}^d$, an $L$-layer KAN defines a function $f$ as follows:

$$f(\boldsymbol{x}) = \boldsymbol{x}^L, \quad \text{where } x_p^l = \sum_{q=1}^{d_{l-1}} \phi_{p,q}^l\big(x_q^{l-1}\big), \quad \boldsymbol{x}^0 = \boldsymbol{x}, \tag{1}$$

where each $\boldsymbol{\phi}^l$ is a matrix of univariate functions of size $d_l \times d_{l-1}$. That is, every entry is a function $\phi_{p,q}^l : \mathbb{R} \to \mathbb{R}$. Conveniently, and analogous to MLPs, the composition of such layers is defined:

$$f(\boldsymbol{x}) = (\boldsymbol{\phi}^L \circ \cdots \circ \boldsymbol{\phi}^1)\boldsymbol{x}, \tag{2}$$

where the operator $\circ$ denotes the application of a layer to its input. To clarify the notation, for a given layer $l$, we define $\big(\phi^l(\boldsymbol{x}^{l-1})\big)_p := \sum_{q=1}^{d_{l-1}} \phi_{p,q}^l(x_q^{l-1})$.

Several parameterizations of these 1D functions are possible (Bozorgasl & Chen, 2024; Zhang et al., 2025). In this work, consistent with the original KAN paper, we adopt a *B-spline*-based parametrization (Schoenberg, 1946), denoted $B(x)$, which represents smooth piecewise polynomial functions over a domain. For a more detailed review of B-splines, see App. B. Specifically, we define the 1D function $\psi(\cdot)$ composing KANs as follows,

$$\psi(x) = w_b\, b(x) + w_s\, B(x); \qquad B(x) = \langle \boldsymbol{c}, \boldsymbol{B}(x) \rangle = \sum_i c_i B_i(x), \tag{3}$$

where $w_b, w_s$ are learnable parameters, $b(x) = \text{silu}(x) = \frac{x}{1+e^{-x}}$, and $B_i(x)$ are pre-defined B-spline basis functions with $c_i$ as the learnable coefficients.

## 3 LEARNING ON KAN PARAMETER SPACES

**Overview.** We begin by analyzing the structure of KANs' parameters and demonstrate that permuting hidden neurons does not alter the underlying function, mirroring the behavior observed in MLPs. Inspired by prior work on weight-space models for MLPs that introduced graph representations (Lim et al., 2023; Kofinas et al., 2024), our approach can be presented as follows: we represent the input KAN as a graph, where nodes correspond to individual neurons and edges represent the connections between them. The learned one-dimensional functions of the KAN are used to define the edge features (details follow). We refer to this construction as the *KAN-graph*; see Figure 1. Importantly, the permutation symmetries—those that leave the underlying KAN function unchanged—correspond to permutations of hidden neurons within the *KAN-graph*, which likewise leave the graph itself unchanged. Thus, we employ a graph neural network-based technique to process the *KAN-graph*, leveraging their inherent equivariance to node permutations.

In what follows, we (1) demonstrate that the permutation symmetries present in MLPs also hold in KANs; (2) present a method for converting KANs into *KAN-graphs*; and (3) introduce WS-KAN a GNN-based architecture for processing *KAN-graphs* and analyze its expressive power.

### 3.1 PERMUTATION SYMMETRIES IN KANS

In a seminal work, Hecht-Nielsen (1990) observed that MLPs exhibit permutation symmetries: re-ordering the neurons within any hidden layer leaves the represented function unchanged. In this subsection, we make the observation that the same symmetry also holds for KANs.

A KAN is fully specified by the collection of one-dimensional functions assigned to each input–output pair in every layer. For convenience, we denote the functions in an $L$-layer KAN by $[\phi^l]_{l \in [L]}$ (where $[L] := \{1, 2, \ldots, L\}$). Given permutation matrices $\boldsymbol{P}_1$ and $\boldsymbol{P}_2$, we define their action on a matrix of univariate functions $\phi$ as,

$$(\boldsymbol{P}_1 \phi \boldsymbol{P}_2)_{p,q} = \phi_{\sigma_1^{-1}(p), \sigma_2(q)}, \tag{4}$$

where $\sigma_1$ and $\sigma_2$ are the permutations associated with $\boldsymbol{P}_1$ and $\boldsymbol{P}_2$, respectively. Intuitively, this corresponds to reordering the rows and columns of $\phi$ according to the given permutations. Below, we formally state the permutation symmetries of KANs.

**Proposition 3.1** (KAN symmetries). *Let $\theta = (\phi^L, \ldots, \phi^1)$ denote the collection of parametric one-dimensional functions composing an L-layer KAN. Consider the group, $G := S_{d_1} \times S_{d_2} \times \cdots \times S_{d_{L-1}}$, the direct product of symmetric groups corresponding to the intermediate dimensions $d_1, \ldots, d_{L-1}$. Let $g = (\boldsymbol{P}_1, \ldots, \boldsymbol{P}_{L-1}) \in G$, where each $\boldsymbol{P}_l$ is the permutation matrix of $\sigma_l \in S_{d_l}$. Define the group action $g \cdot \theta = \theta'$ with $\theta' = (\phi'^L, \ldots, \phi'^1)$ given by,*

$$\phi'^1 = \boldsymbol{P}_1^\top \phi^1, \qquad \phi'^l = \boldsymbol{P}_L^\top \phi^l \boldsymbol{P}_{l-1}, \ \forall l = 2, \ldots, L-1, \qquad \phi'^L = \phi^L \boldsymbol{P}_{L-1}.$$

*Then, $f_\theta(\boldsymbol{x}) = f_{\theta'}(\boldsymbol{x})$ for all $\boldsymbol{x}$.*

A short proof is given in App. E for the case of a single hidden-layer KAN, and the extension to multiple hidden layers follows naturally.

Intuitively, such permutations correspond to reordering nodes within hidden layers. As an example, Figure 2 illustrates a single–hidden-layer KAN, where the permutation matrix $\boldsymbol{P}$ corresponds to $(1, 3)$. In other words, the first and third nodes are mapped to one another, while the second remains unchanged. Importantly, these permutation symmetries hold independently of the chosen parametrization of the 1D functions.

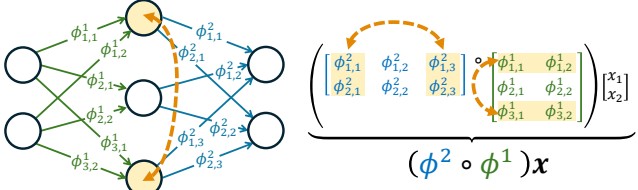

Figure 2: Hidden neuron permutation symmetries in KANs.

In the next subsection, we introduce the *KAN-graph*, a graph representation of KANs that (i) compactly encodes their structure, and (ii) remains invariant under neuron permutations that do not alter the function computed by the KAN.

### 3.2 KAN-GRAPH

The main idea behind the definition of the KAN-graph is relatively natural: as illustrated in Figure 1, nodes represent the KAN's neurons and edges should represent the univariate functions. In this subsection, we formalize the construction by explicitly defining the nodes, edges, and edge features derived from those univariate functions.

We define the *KAN-graph* as a directed graph $G = (V, E)$ where $V$ represents the set of neurons, $E$ represents the set of edges, and $\boldsymbol{v} \in \mathbb{R}^{n \times d_V}, \boldsymbol{e} \in \mathbb{R}^{m \times d_E}$ are their corresponding features, respectively. Here, $n, m$ denote the total number of nodes and edges in the graph, respectively. To clarify, in Figure 2(left), we have $n = 7$ nodes, $m = 12$ edges, and the corresponding adjacency matrix is visualized inset.[1]

| $\leftarrow d_0 \rightarrow$ | $\leftarrow\ \ \ d_1\ \ \ \rightarrow$ | | $\leftarrow d_2 \rightarrow$ | |
|---|---|---|---|---|
| $\mathbf{0}$ | $\begin{matrix}\phi_{1,1}^1 & \phi_{2,1}^1 & \phi_{3,1}^1 \\ \phi_{1,2}^1 & \phi_{2,2}^1 & \phi_{3,2}^1\end{matrix}$ | | $\mathbf{0}$ | $d_0$ |
| $\mathbf{0}$ | $\mathbf{0}$ | | $\begin{matrix}\phi_{1,1}^2 & \phi_{1,2}^2 \\ \phi_{2,1}^2 & \phi_{2,2}^2 \\ \phi_{3,1}^2 & \phi_{3,2}^2\end{matrix}$ | $d_1$ |
| $\mathbf{0}$ | $\mathbf{0}$ | | $\mathbf{0}$ | $d_2$ |

[1]The adjacency matrix of a given KAN-graph is extremely sparse: nonzeros appear only in the first super-diagonal blocks, specifically between blocks $l$ and $l+1$ for each $l \in [L]$.

**Univariate functions as edge features in the *KAN-graph*.** To fully define the *KAN-graph*, we must specify how to incorporate the explicit parametrization of $[\boldsymbol{\phi}^l]_{l\in[L]}$ characterizing the KAN into the KAN-graph. In this work, we focus on the parametrization introduced in the original KAN paper (Liu et al., 2025), wherein the learnable univariate functions are based on B-splines; as per Eq. (3).

In this case, the learnable parameters composing those univariate functions can be conveniently 'collected' into a vector $\tilde{\boldsymbol{\phi}}_{p,q}^l := [w_{b;p,q}^l,\ w_{s;p,q}^l,\ \boldsymbol{c}_{p,q}^l]$. Thus, we define, for a layer $l$, an input node (or neuron) $p$, and an output node $q$, the following edge feature,

$$\boldsymbol{e}_{p,q}^l = \tilde{\boldsymbol{\phi}}_{p,q}^l := [w_{b;p,q}^l,\ w_{s;p,q}^l,\ \boldsymbol{c}_{p,q}^l], \tag{5}$$

which elegantly collects the learnable parameters of the one-dimensional function $\phi_{p,q}^l$.

### 3.3 Learning on the *KAN-graph*

To learn on the *KAN-graph*, we adopt a general message-passing framework motivated by Gilmer et al. (2017), as follows (the letters a–d denote the order of execution),

$$a)\ \boldsymbol{v}_i^{\mathrm{F}} \leftarrow \mathrm{MLP}_v^{(2;\mathrm{F})}\Big(\boldsymbol{v}_i, \sum_{j:\,\boldsymbol{e}(i,j)\in E} \mathrm{MLP}_v^{(1;\mathrm{F})}(\boldsymbol{v}_j, \boldsymbol{e}_{(i,j)})\Big);\ b)\ \boldsymbol{v}_i^{\mathrm{B}} \leftarrow \mathrm{MLP}_v^{(2;\mathrm{B})}\Big(\boldsymbol{v}_i, \sum_{j:\,\boldsymbol{e}(i,j)\in E^T} \mathrm{MLP}_v^{(1;\mathrm{B})}(\boldsymbol{v}_j, \boldsymbol{e}_{(i,j)})\Big);$$

$$c)\ \boldsymbol{e}_{(i,j)} \leftarrow \mathrm{MLP}_e(\boldsymbol{v}_i, \boldsymbol{v}_j, \boldsymbol{e}_{(i,j)});\qquad\qquad d)\ \boldsymbol{v}_i \leftarrow \mathrm{MLP}_v^{(3)}(\boldsymbol{v}_i, \boldsymbol{v}_i^{\mathrm{F}}, \boldsymbol{v}_i^{\mathrm{B}}).$$

Intuitively, node features are updated by aggregating information from both their outgoing and incoming neighbors. Edge features, in turn, are refined based on the states of their endpoints as well as their own current representation. Each node's representation is then updated by combining its intrinsic features with the forward- and backward-aggregated information. We note that although the computational graph of KANs is inherently directed, with edges pointing from one layer to the next, we explicitly perform bidirectional message passing – propagating information not only forward but also in reverse – as this dual flow was found to enhance performance.

**Positional encodings (PE) as additional edge and node features.** We follow prior work (Kofinas et al., 2024; Lim et al., 2024) and augment each node and edge of the KAN-graph with positional embeddings that indicate their position in the computation flow. This breaks potential artificial symmetries that may arise in the KAN-graph. Specifically, all nodes within the same intermediate layer share a common positional embedding. By contrast, input and output nodes are assigned distinct embeddings, since permutations of these nodes generally alter the network's function. For edges, we assign a unique identifier to each one, associated with its input and output nodes.

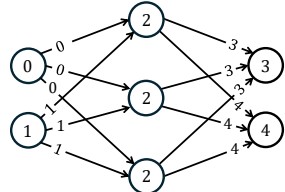

Figure 3: PE.

Figure 3 depicts a possible assignment of positional encoding (via simple integers) to both nodes and edges for the running example of Figure 2(left). Intuitively, permuting neurons within a hidden layer leaves the feature-augmented graph (i.e., its adjacency matrix) unchanged. In contrast, permuting a node from the first layer with one from the last *does* alter the graph.

## 4 Expressive power of WS-KAN

While there are multiple ways to design a WS architecture for processing KANs, we adopted *one* specific approach, which stems from equivariance principles, and models the KAN as a graph. However, a question arises: *is this the right design choice?*

In general, imposing group-equivariance constraints might reduce a model's expressive power (Maron et al., 2019; Xu et al., 2019; Morris et al., 2019). In the weight-space literature, this expressive power is often analyzed by demonstrating that the architecture can simulate (i.e., approximate) the forward pass of a given input model. For example, Lim et al. (2024); Navon et al. (2023) establish such results in the context of processing standard neural networks. Such approximation results typically serve as an intermediate step toward proving stronger results, such as functional approximation theorems (Navon et al., 2023).

In this section, we show that WS-KAN can simulate a forward pass for a given input KAN. We begin by showing that there exists a set of weights for a single-hidden-layer MLP that can approximate any univariate function based on B-splines (under mild assumptions), as per Eq. (3), to arbitrary precision. We then use this result to prove our main result: WS-KAN can simulate the forward pass of the original KAN. All proofs are provided in App. E.

**Lemma 4.1** (MLP as an approximation of the univariate functions composing the KAN). *Let $\mathcal{B}$ denote the family of cardinal B-splines of degree $k$, defined on a fixed grid $G$ over the domain $[a, b]$, and parameterized by coefficients $\boldsymbol{c} = [c_0, \ldots, c_{G+k-1}] \in \mathcal{C} \subset \mathbb{R}^{G+k}$, where the set of admissible coefficients lies in a compact domain $\mathcal{C}$. Consider the function $\psi : \mathbb{R} \to \mathbb{R}$ defined as $\psi(x) = w_b\, b(x) + w_s\, B(x)$, where $w_b, w_s \in \mathcal{W} \subset \mathbb{R}$ for some compact set $\mathcal{W}$. $b(x)$ denotes the silo function, and $B(x)$ denotes the B-spline. Then, for any $\varepsilon > 0$, and for any compact domain $\mathcal{X}$ there exists a set of weights for a multilayer perceptron $\mathtt{MLP} : \mathbb{R}^{G+k+3} \to \mathbb{R}$ such that,*

$$\sup_{x \in \mathcal{X}} \left| \mathtt{MLP}(x, w_s, w_b, \boldsymbol{c}) - \psi(x) \right| < \varepsilon.$$

Lemma 4.1 essentially states that, under the assumptions above, there exists an $\mathtt{MLP}$ that effectively computes the univariate functions composing our input KANs, over any chosen compact domain $\mathcal{X}$. Consequently, we show that under mild conditions, WS-KAN can approximate a forward pass of the input KAN model.

**Proposition 4.2** (WS-KAN can simulate the forward pass of KANs). *Let $f_\theta$ be a given KAN architecture, defined over an input domain $[a, b]^n$, where each univariate function is represented by a B-spline from the family $\mathcal{B}$. Let $G$ be its* KAN-graph*, where the nodes in the first layer are enhanced with the input $\boldsymbol{x} \in [a, b]^n$. For every $\varepsilon > 0$, there exists a* WS-KAN *such that,*

$$\sup_{\boldsymbol{x} \in [a,b]^n} \left| \textit{WS-KAN}(G) - f_\theta(\boldsymbol{x}) \right| < \varepsilon.$$

Importantly, Proposition 4.2 holds for any choice of parameters defining the KANs' univariate functions (Eq. (3)). Below we present the proof idea, while the full proof can be found in App. E.

*Proof idea.* A KAN computation can be viewed as a composition of $L$ continuous functions (layers). Using Lemma 4.1, we show that WS-KAN can approximate each individual layer to arbitrary precision. Then, by applying standard techniques (e.g., ideas from Lemma 6 of Lim et al. 2022) and the message passing mechanism, we construct approximations layer by layer, ensuring that the overall WS-KAN output remains arbitrarily close to that of the original KAN. □

## 5 EXPERIMENTS

While various "model zoos" (e.g., Schürholt et al., 2022e) exist for benchmarking weight-space models in conventional neural networks, no comparable resource has yet been developed for KANs. Nor are there established baselines against which WS-KAN can be evaluated. To address this gap, we take inspiration from tasks and baselines explored in the weight-space literature for conventional neural networks and construct several families of model zoos of trained KANs. These zoos are designed to capture both invariant and equivariant tasks and are built from five datasets: MNIST (LeCun et al., 1998), Fashion-MNIST (F-MNIST; Xiao et al. 2017), Kuzushiji-MNIST (K-MNIST; Clanuwat et al. 2018), CIFAR10 (Krizhevsky et al., 2009), and a synthetic dataset that we designed inspired by the one in Navon et al. (2023).

We focus on two invariant problems—INR classification (Section 5.1) and accuracy prediction (Section 5.2)—and one equivariant task—pruning, where the goal is to predict a pruning mask directly from the KAN's weights (Section 5.3). We also evaluate WS-KAN's ability to generalize to KAN architectures unseen during training (Section 5.1.1). For each model in the zoo, we also construct the corresponding KAN-graph. We benchmark WS-KAN against the following baselines.

***Standard baselines:*** (1) **MLP**: A simple multilayer perceptron applied to a vectorized representation of the KAN's parameters. (2) **MLP + Aug.**: The same MLP as above, but trained with permutation augmentation, i.e., randomly permuting the input KAN in ways that preserve its underlying function. (3) **MLP + Align.**: Inspired by alignment techniques for MLPs (Ainsworth et al., 2022), where parameters are reordered to maximize model similarity, we extend this idea to KANs. The main challenge is that KAN parameters are functions rather than scalars, and therefore alignment

requires defining distances between functions. Full details are in App. D. We consider as a baseline an MLP applied to aligned KANs. (4) **DMC** (Eilertsen et al., 2020): a convolutional layer applied to the (vectorized) model parameters. *Ablation baselines:* We introduce two additional baselines to ablate our architectural design choice. (5) **DS (DeepSets)**: A DeepSets (Zaheer et al., 2017) architecture applied to the graph's edge features. Importantly, while this baseline is invariant to KAN permutation symmetries, it is also invariant to many more permutations that do not correspond to these symmetries, and completely neglects the graph topology. (6) **SetTrans**: Similar to (5), but employing a transformer architecture (Vaswani et al., 2017) over the set of edges. We note that this baseline is only feasible for relatively small input KANs, since the attention matrix scales quadratically with the number of neurons.[2] Thus, **SetTrans** is reported only when feasible. Additionally, we ablate key design choices in WS-KAN, including positional encoding and bidirectional message passing; see App. C.4 for details.

In the following sections, we provide additional details on the tasks considered, describe the construction of the KAN training datasets, and present our results. For each experiment, we report mean performance over three seeds, with error bars for standard deviation. All test results were obtained by optimizing for validation performance. Additional experimental details, including the hyperparameter grid, implementation notes, and extended results, are available in App. C[3].

## 5.1 INR CLASSIFICATION

First, we evaluate WS-KAN on INR (Mescheder et al. 2019; Sitzmann et al. 2020) classification.

**What is an INR?** For a given image, an INR learns a mapping from any input coordinate to the corresponding grayscale (or RGB) value of that coordinate in the image. See inset (top). At the bottom inset, we show example reconstructions produced by KAN-based INRs over CIFAR10, F-MNIST, MNIST – left corresponds to ground truth image, and right corresponds to the reconstructed one.

**Setup and dataset construction.** The setup is illustrated inset. For this task, we convert the following datasets to KAN-based INRs: Sine waves (a synthetic dataset), MNIST, F-MNIST, and CIFAR10. Taking MNIST as an example, the key idea is that

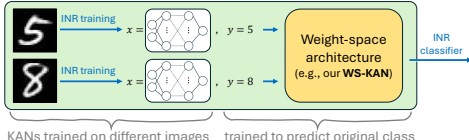

for *each* image in the dataset, we train an *independent* KAN-based INR to 'reconstruct' it. Once this zoo of INRs is constructed, we train the weight-space model under study (e.g., WS-KAN) to classify the digit, using as input the parameters of the KAN-based INR (or the *KAN-graph* when WS-KAN is tested) rather than the raw pixel data. See App. C.1 for dataset split and details.

**Results.** Table 1 reports the accuracy of predicting the class (e.g., the digit in MNIST) from the weights of the KAN-based INR. WS-KAN outperforms all baselines by a large margin. SetTrans ranks second, suggesting that explicitly accounting for symmetries is advantageous, albeit suboptimal, as the symmetries it captures are broader than the KAN permutation symmetries. Finally, we observe that "*MLP + Align. > MLP + Aug. > MLP*", aligning with intuition and validating the effectiveness of our alignment technique. Results over the synthetic dataset are provided in App. C.1.3.

Table 1: **INR classification accuracy.**

| Method | MNIST | F-MNIST | CIFAR-10 |
|---|---|---|---|
| MLP | 34.1±0.1 | 41.3±0.3 | 16.8±0.1 |
| MLP + Aug. | 62.7±1.1 | 63.0±0.3 | 28.2±0.7 |
| MLP + Align. | 81.0±0.1 | 73.6±0.2 | 30.0±0.2 |
| DMC | 73.4±3.0 | 73.1±1.0 | 33.0±0.7 |
| DS (Ours) | 59.1±3.3 | 65.9±0.8 | 23.2±3.9 |
| SetTrans (Ours) | 87.5±0.8 | 80.2±0.1 | 34.3±0.7 |
| WS-KAN (Ours) | **94.3**±0.5 | **84.6**±0.6 | **42.2**±0.8 |

---

[2]In practice, we found this approach computationally expensive—training a single epoch could take up to one hour in some experiments—making it significantly slower than WS-KAN and other baselines.

[3]Our code, including model zoo construction, is available at https://github.com/BarSGuy/KAN-Graph-Metanetwork.

### 5.1.1 OUT-OF-DISTRIBUTION GENERALIZATION TO WIDER KANS

Since WS-KAN operates on KAN-graphs via a GNN-based architecture, it can naturally be applied to KAN-graphs of varying topologies, e.g., differing in hidden-layer width or number of layers. Hence, here we evaluate WS-KAN's ability to generalize to KAN architectures larger than those seen during training, focusing on the INR classification task over the MNIST and F-MNIST datasets. Our KAN-based INRs from Section 5.1 share the topology $[2 \to h \to h \to 10]$, where $h$ denotes the hidden-layer width, with $h = 32$. Here, we take those WS-KAN's trained on the $h = 32$ KANs and evaluate it on progressively wider KAN architectures with $h \in \{48, 64, 80, 96\}$, none of which were seen during training.

**Results.** Table 2 reports the out-of-distribution (OOD) accuracy of WS-KAN. We observe promising OOD generalization, which, as expected, degrades as $h$ increases and the distribution shift from the training setting ($h = 32$) grows.

Table 2: **OOD generalization for INR classification.** Test accuracies of WS-KAN on the INR classification task when trained on KANs with hidden width $h = 32$ and evaluated on wider, previously unseen architectures ($h \in \{48, 64, 80, 96\}$). All architectures follow the topology $[2 \to h \to h \to 10]$. The  blue column denotes the in-distribution setting , while  red columns denote out-of-distribution setting .

| Dataset | $h = 32$ | $h = 48$ | $h = 64$ | $h = 80$ | $h = 96$ |
|---|---|---|---|---|---|
| MNIST | $94.3 \pm 0.5$ | $91.4 \pm 0.5$ | $81.0 \pm 3.2$ | $67.0 \pm 4.3$ | $57.1 \pm 6.1$ |
| F-MNIST | $84.6 \pm 0.6$ | $84.6 \pm 0.6$ | $84.3 \pm 0.7$ | $83.3 \pm 0.8$ | $82.2 \pm 0.7$ |

## 5.2 ACCURACY PREDICTION

Here, we consider the task of predicting the accuracy of a given KAN, based on its parameters.

**Setup and dataset construction.** We experiment on MNIST, F-MNIST, and K-MNIST. Over each dataset, we train 4000 KAN models with a 3000/500/500 train/validation/test split. We observed that different KANs trained on these datasets yield similar accuracies. Thus, to make predicting accuracy from parameters more challenging, we introduce label noise: for each KAN, we randomly sample portions of the training data and shuffle their labels. As shown inset, this results in a diverse set of trained KANs with varying test accuracies (results for other datasets provided in Figure 9 in App. C). The training pipeline is illustrated in Figure 10 in App. C.2.

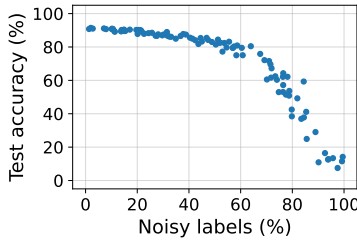

**Results.** To test how well WS-KAN predicts the test accuracies, we follow Lim et al. (2024) and report two metrics: Mean Squared Error (MSE) and R-squared ($R^2$). The results, summarized in Table 3, exhibit the same trend as in INR classification: WS-KAN consistently achieves the best performance across all datasets. The ordering of baselines remains unchanged, with our ablation baseline (DS) ranking second, followed by *MLP + Alignment* as the next most effective approach.

Table 3: **Accuracy prediction**. Comparison of MSE and $R^2$ across datasets.

| Method | **MSE** (lower is better; ↓) [$\times 10^3$] | | | **R$^2$** (higher is better; ↑) [$\times 10^2$] | | |
|---|---|---|---|---|---|---|
| | MNIST | F-MNIST | K-MNIST | MNIST | F-MNIST | K-MNIST |
| MLP | $14.58 \pm 0.02$ | $11.55 \pm 0.12$ | $6.68 \pm 0.16$ | $76.99 \pm 0.04$ | $69.59 \pm 0.33$ | $80.13 \pm 0.49$ |
| MLP + Aug. | $10.41 \pm 0.24$ | $8.86 \pm 1.47$ | $5.33 \pm 0.19$ | $83.56 \pm 0.38$ | $76.66 \pm 3.88$ | $84.15 \pm 0.55$ |
| MLP + Align. | $5.26 \pm 0.09$ | $6.32 \pm 0.15$ | $3.33 \pm 0.10$ | $91.70 \pm 0.15$ | $83.37 \pm 0.41$ | $90.11 \pm 0.31$ |
| DMC | $7.00 \pm 0.06$ | $6.46 \pm 0.32$ | $3.27 \pm 0.05$ | $88.95 \pm 0.09$ | $82.99 \pm 0.85$ | $90.29 \pm 0.14$ |
| DS (Ours) | $\mathbf{3.29} \pm 0.12$ | $3.90 \pm 0.04$ | $2.00 \pm 0.11$ | $\mathbf{94.81} \pm 0.18$ | $89.73 \pm 0.11$ | $94.07 \pm 0.32$ |
| WS-KAN (Ours) | $\mathbf{3.29} \pm 0.17$ | $\mathbf{2.94} \pm 0.13$ | $\mathbf{1.45} \pm 0.08$ | $\mathbf{94.81} \pm 0.27$ | $\mathbf{92.27} \pm 0.35$ | $\mathbf{95.69} \pm 0.24$ |

## 5.3 PRUNING MASK PREDICTION

In this section, we tackle the challenging equivariant task of *network pruning* for KANs, aiming to discard a subset of weights without significantly degrading performance.

**Motivation.** Most pruning methods are data-driven, requiring large amounts of data to determine which parameters to remove. For instance, activation-based approaches rely on recorded activation values, while gradient-based methods depend on training loss gradients. In contrast, pruning a model using only a simple forward pass, as enabled by WS-KAN (demonstrated below), is especially valuable, as it avoids repeated, data-intensive passes.

**Setup.** Here, we use the same datasets as in Section 5.2. Supervision is obtained by applying a data-driven edge-based pruning algorithm for KANs[4], which we denote as *Oracle-prune*. The core idea behind this algorithm is straightforward: it removes edges whose average activation values, computed from training data, fall below a predefined threshold (set to 0.01 in all experiments). We refer to this pruning method as *Oracle-pruning*, as it is our supervision. The oracle pruning algorithm outputs a binary mask, where edges marked with 0 are pruned and those marked with 1 are retained. The task of interest is to predict this mask, see Figure 11 in App. C.3 for the training pipeline. Crucially, this task is equivariant: prediction is made for each individual edge of the KAN, rather than as a single prediction for the entire network. We are interested in evaluating two aspects: **(i)** how accurately WS-KAN predicts the pruning mask, and **(ii)** whether using the mask generated by WS-KAN leads to effective downstream pruning performance.

Table 4: **Pruning mask prediction.**

| | Accuracy (↑, %) | | | ROC-AUC (↑, %) | | |
|---|---|---|---|---|---|---|
| **Method** | MNIST | F-MNIST | K-MNIST | MNIST | F-MNIST | K-MNIST |
| MLP | $93.10{\pm}{<}0.01$ | $96.65{\pm}{<}0.01$ | $91.39{\pm}{<}0.01$ | $87.12{\pm}0.04$ | $84.92{\pm}0.38$ | $75.32{\pm}0.02$ |
| MLP + Aug. | $93.29{\pm}{<}0.01$ | $96.65{\pm}{<}0.01$ | $91.39{\pm}{<}0.01$ | $91.36{\pm}0.12$ | $86.21{\pm}0.09$ | $74.89{\pm}0.01$ |
| MLP + Align. | $93.57{\pm}0.01$ | $96.64{\pm}{<}0.01$ | $91.52{\pm}0.01$ | $93.00{\pm}0.03$ | $91.66{\pm}0.10$ | $82.76{\pm}0.05$ |
| DMC | $93.07{\pm}{<}0.01$ | $96.59{\pm}{<}0.01$ | $91.39{\pm}{<}0.01$ | $84.27{\pm}0.06$ | $84.39{\pm}0.01$ | $75.06{\pm}0.02$ |
| DS (Ours) | $94.34{\pm}0.02$ | $96.90{\pm}0.09$ | $94.34{\pm}0.02$ | $95.45{\pm}0.05$ | $95.81{\pm}0.04$ | $95.45{\pm}0.05$ |
| WS-KAN (Ours) | $\mathbf{97.93}{\pm}0.19$ | $\mathbf{98.93}{\pm}0.05$ | $\mathbf{97.72}{\pm}0.14$ | $\mathbf{99.54}{\pm}0.01$ | $\mathbf{99.72}{\pm}0.02$ | $\mathbf{99.46}{\pm}0.09$ |

**Results (i).** We evaluate mask prediction as a binary classification task using the metrics ROC-AUC and Accuracy. Results are provided in Table 4. WS-KAN consistently outperforms all baselines across all datasets and evaluation metrics. Again, the DS approach emerges as the second best overall. Notably, the hierarchy among the MLP variants mirrors our earlier findings, further supporting the effectiveness of our alignment strategy.

**Results for downstream pruning performance (ii).** To properly evaluate downstream pruning, it is not enough to only report the downstream accuracy achieved by applying a mask. A trivial mask that leaves all weights untouched would naturally yield high accuracy, but would provide no practical benefit. A meaningful mask must therefore strike a balance: it should maintain strong downstream accuracy while also inducing sparsity in the model.

We now evaluate how well WS-KAN achieves this trade-off, and present two complementary plots. In Figure 5a, we report the downstream accuracy (y-axis) achieved by pruned models across different noise levels in the training labels (x-axis, binned in 20% non-overlapping intervals). In Figure 5b, we show the corresponding fraction of weights retained by each method. Together, these plots illustrate both the accuracy–sparsity trade-off and how each method compares against the oracle-pruning baseline. It is clear that WS-KAN most closely follows the accuracy–sparsity trade-off of the *oracle-pruning* technique. Moreover, DS consistently ranks second best, whereas all other approaches perform poorly. The only partial exception is MLP + Alignment, which shows some utility but still falls significantly behind both DS and WS-KAN. Additional downstream pruning results are available in App. C. Importantly, we observe that WS-KAN offers a significant timing advantage over *Oracle Prune*, being up to five orders of magnitude faster (Figure 5c).

---

[4]This (off-the-shelf) pruning algorithm is found in `https://github.com/KindXiaoming/pykan`

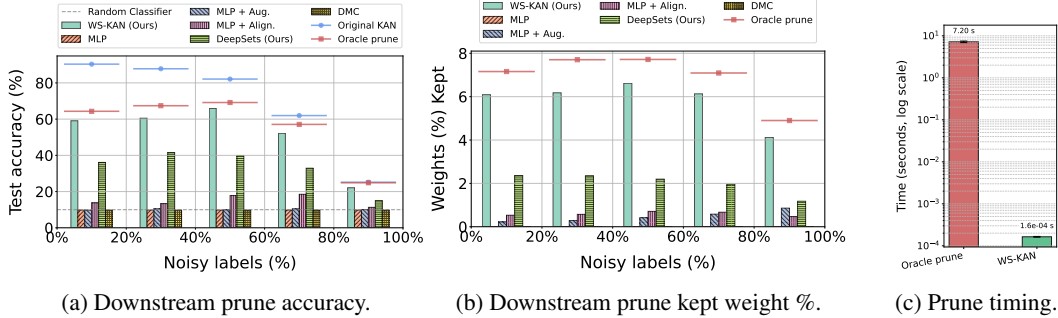

(a) Downstream prune accuracy.     (b) Downstream prune kept weight %.     (c) Prune timing.

Figure 5: **Downstream pruning performance across methods over KANs trained on MNIST.** We report: (i) **Test accuracy**: the downstream accuracy of pruned networks, averaged over non-overlapping bins of 20%, to highlight the relative effectiveness of pruning strategies under varying noise levels – Figure 5a; (ii) **Kept weights**: the percentage of weights retained after pruning, averaged over the same bins as in (i) – Figure 5b; and (iii) **Pruning time** (↓): the computational cost comparison (log scale) in seconds, between WS-KAN and Oracle prune – Figure 5c, low is better.

**Discussion.** WS-KAN consistently provides the most effective WS technique for learning over KANs. Importantly, our ablation baselines (DS/SetTrans) are generally second-best. This underscores the value of incorporating the structural properties of KANs, since even a suboptimal integration (as per DS/SetTrans) still proves beneficial to some extent. Although our alignment method (see App. D) does not outperform structure-aware approaches, it remains a strong alternative. Across nearly all tasks/datasets/metrics combinations, we consistently observe the ordering: "*MLP + Align.* > *MLP + Aug.* > *MLP*". Finally, we note that WS-KAN is computationally efficient, with complexity scaling linearly in the number of edges of the KAN-graph; see App. C.5 for a detailed complexity and runtime analysis.

## 6    CONCLUSIONS

We addressed the development of weight-space models for KANs by performing a symmetry analysis that guided the design of our WS-KAN architecture. We also built comprehensive model zoos to support future research and evaluation of weight-space models for KANs.

Several promising future directions remain open. While we have shown that WS-KAN can generalize to wider KAN architectures unseen during training (Section 5.1.1), extending this to deeper architectures and broader topology variations remains an exciting avenue. Beyond this, the availability of weight-space models for both KANs and MLPs naturally invites the study of transformations between the two, enabling us to leverage their complementary strengths. Many analytical tools are well established for MLPs; by converting a KAN into an equivalent MLP (while preserving its function), these tools can be used to study the original KAN. Conversely, converting an MLP into a KAN allows us to exploit the interpretability of KANs (Barašin et al., 2024), offering new insights into the MLP. Finally, evaluating WS-KAN's generalization to other KANs – e.g., CNN-KANs – would be an interesting direction.

## ACKNOWLEDGMENTS

G.B. is supported by the Jacobs Qualcomm PhD Fellowship. HM is supported by the Israel Science Foundation through a personal grant (ISF 264/23) and an equipment grant (ISF 532/23), and by the Career Advancement Chairs in Artificial Intelligence – Schmidt Futures.

## REPRODUCIBILITY STATEMENT

Our code is available at https://github.com/BarSGuy/KAN-Graph-Metanetwork. All implementation details required to replicate the results and evaluations are provided in App. C.

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

## A  APPENDIX ROADMAP

This appendix gathers the mathematical background, full experimental details, the alignment procedure, and proofs that support the main text. See guide below.

- **App. B (B-splines).** Reviews uniform B-splines and clarifies how they parameterize the univariate functions in KANs.

- **App. C (Extended Experimental Section).** Provides all details to create the model zoos for each task and reproduce all results:
    - **App. C.1** – INR classification.
    - **App. C.2** – Accuracy prediction.
    - **App. C.3** – Pruning mask prediction.
    - **App. C.4** – Ablation: positional encoding and bidirectional message passing.
    - **App. C.5** – Complexity and runtime analysis.

- **App. D (Aligning Kolmogorov-Arnold Networks).** Here we extend the alignment ideas presented in Ainsworth et al. (2022) to KANs.
    - **App. D.1 (Aligning a model zoo or a dataset).** Here we describe how our alignment procedure discussed in App. D can be applied to an entire dataset of trained KANs.

- **App. E (Proofs).** Collects all formal propositions and proofs.

- **App. F (Large Language Model (LLM) Usage).** Here we provide details on how we used LLMs.

## B  B-SPLINES

B-splines are smooth piecewise polynomial functions over a domain $[a, b]$, defined using a knot vector $\mathbf{T}$. Specifically, a degree-$k$ B-spline with $G$ grid points is expressed as:

$$B(x) = \langle \boldsymbol{c}, \boldsymbol{B}(x) \rangle = \sum_{i=0}^{G+k-1} c_i B_i(x), \qquad (6)$$

where $B_i(x)$ are basis functions and $c_i$ are the learnable parameters; see the inset[5] illustration for the basis functions corresponding to $G = 5$ and $k = 3$. The knot vector $\mathbf{T} = [t_{-k}, \ldots, t_0, t_1, \ldots, t_G, \ldots, t_{G+k}]$ is larger than the domain itself, and determines where and how the basis functions are defined. The basis functions are defined recursively using the Cox–de Boor formula (De Boor, 1978); the explicit expression is provided below. For uniform B-splines, of which we focus on in this paper, the knots are equally spaced with $t_0 = a$, $t_G = b$, and spacing $\Delta t = t_i - t_{i-1}$.

**Cox–de Boor Formula** The Cox–de Boor recursion De Boor (1978) defines the B-spline basis functions as follows:

$$N_{i,k'}(x) = \begin{cases} 1, & \text{if } t_i \leq x < t_{i+1}, \\ 0, & \text{otherwise}, \end{cases} \quad \text{for } k' = 0,$$

$$N_{i,k'}(x) = \frac{x - t_i}{t_{i+k'} - t_i} N_{i,k'-1}(x) + \frac{t_{i+k'+1} - x}{t_{i+k'+1} - t_{i+1}} N_{i+1,k'-1}(x), \quad \text{for } k' > 0.$$

We denote

$$B_i(x) := N_{i-k,k}(x).$$

In the uniform case, the denominators simplify to $k'\Delta t$.

---

[5] The figure is taken from Liu et al. (2025).

## C  EXTENDED EXPERIMENTAL SECTION

The implementation of WS-KAN was carried out using PyTorch (Paszke et al., 2019) and Py-Torch Geometric (Fey & Lenssen, 2019), which are distributed under the BSD and MIT licenses, respectively. Hyperparameter optimization was conducted with the Weight and Biases framework (Biewald, 2020). Below we provide the additional details necessary to reproduce our experiments. These include instructions on how we constructed the model zoos for each task, how we trained both WS-KAN and the baseline models, as well as the hyperparameter grids and other relevant configurations. Importantly, for all WS models considered we employed the exact same data split and hyperparameter grid search. For the specific procedure used to align a dataset of trained KANs, please refer to App. D.1.

**Optimizer and schedulers.** For all considered datasets and tasks for WS models considered, we use the AdamW optimizer Loshchilov & Hutter (2019) in combination with a linear learning rate scheduler, incorporating a warm-up phase over the first 100 of training steps.

All KANs we have trained for constructing the various model zoos use the `fit` function from the PyKAN library[6].

### C.1  INR CLASSIFICATION: EXTENDED SECTION

### C.1.1  MODEL ZOO – INR CLASSIFICATION

**Constructing the synthetic 2D sine wave INR dataset.** We start by sampling a frequency vector uniformly at random from the range $\boldsymbol{w} \in [0.5, 10]^2$. Each sample defines a 2D sine wave of the form,

$$g(\boldsymbol{x}) = \sin(\boldsymbol{w} \cdot \boldsymbol{x}),$$

where $\boldsymbol{x}$ is the input and $g(\boldsymbol{x})$ is the target output. To learn this mapping, we train a KAN-based implicit neural representation (INR) with architecture depth/width configuration of $[2, 32, 32, 1]$. We train $1,000$ independent models, using a split of 800/100/100 for training, validation, and testing, respectively. Training is performed with a batch size of 128, and the learning rate is fixed at $0.01$, for $1,000$ epochs. The univariate functions in the KAN is parameterized via B-splines with $G = 30$ and $k = 3$. See example of a KAN-based INR for a samples sine function in Figure 6.

**Constructing INRs for MNIST, F-MNIST, CIFAR10.** For each dataset, we train a KAN-based implicit neural representation (INR) model that maps pixel coordinates to their corresponding intensity values—grayscale for MNIST and F-MNIST, and RGB for CIFAR10. The network architecture is configured as $[2, 32, 32, 1]$ for MNIST and F-MNIST, and $[2, 32, 32, 3]$ for CIFAR10 to account for its three color channels. For all datasets, we adopt a batch size of 128 and train for $1,000$ epochs using a fixed learning rate of $0.01$. The univariate functions in the KAN is parameterized via B-splines with $G = 10$ and $k = 3$. An illustration of a KAN-based INR is provided in Figure 7.

---

[6] https://github.com/KindXiaoming/pykan

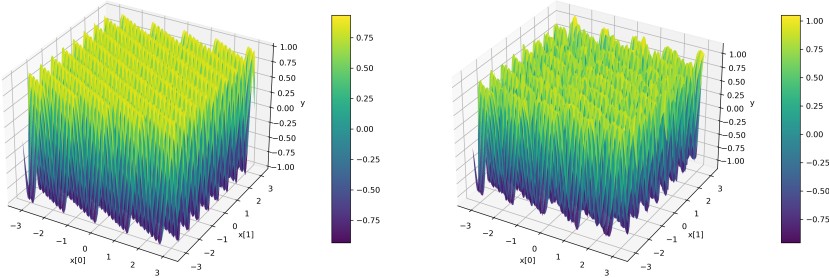

Figure 6: Example of a KAN-based INR applied to the synthetic 2D sine wave dataset. The left panel shows the ground truth, and the right panel shows the reconstructed result.

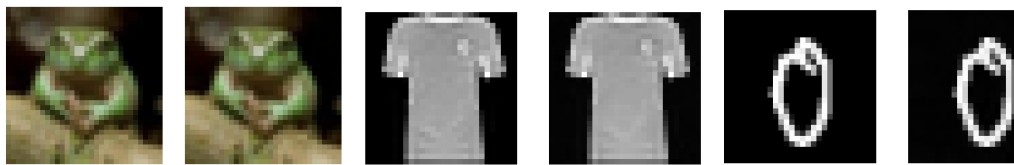

Figure 7: Reconstructions from INR on CIFAR-10, Fashion-MNIST, and MNIST. All PSNRs are more than 40.

### C.1.2 INR CLASSIFICATION - ADDITIONAL DETAILS

**Hyperparameters.** For all datasets considered (except the sine wave dataset), we trained both WS-KAN and the baseline models on a randomly sampled subset of 10,000 trained KANs, with an additional 5,000 reserved for validation. The KANs used for testing are those that were trained on the original dataset's test images. For each considerd WS model, we used 4 layers, a hidden dimension of 128, batch size of 128, weight decay of 0.01, dropout of 0.2, and 300 epochs. For the learning rate, we performed a search over the set $\{0.001, 0.0001\}$.

For the sine wave dataset, we made the following adjustments: training was conducted for 200 epochs with a fixed learning rate of 0.001. Additionally, we experimented with varying training set sizes, using subsets of $\{100, 200, 500, 800\}$ from the available 800 samples.

### C.1.3 RESULTS ON SYNTHETIC DATASET

**Synthetic dataset: 2D Sine waves.** We constructed a dataset of KAN-based INRs for 2D sine waves – see App. C.1.1 for more details on this dataset construction. The task is to predict the frequency $w$ of a given test KAN-based INR, via its parameters. To evaluate the generalization capabilities of the architectures on this synthetic dataset, we repeat the experiment while varying the number of training examples (INRs). As shown in Figure 8, WS-KAN consistently outperforms all baseline methods, even when trained with only a small number of examples.

### C.2 ACCURACY PREDICTION – EXTENDED SECTION

The weight-space training pipeline for the task of accuracy prediction is illustrated in Figure 10.

### C.2.1 MODEL ZOO – ACCURACY PREDICITON

To construct the model zoos for the task of accuracy prediction, we used MNIST, F-MNIST, and K-MNIST. For each dataset, we trained 4,000 KAN models with a 3000/500/500 split for training,

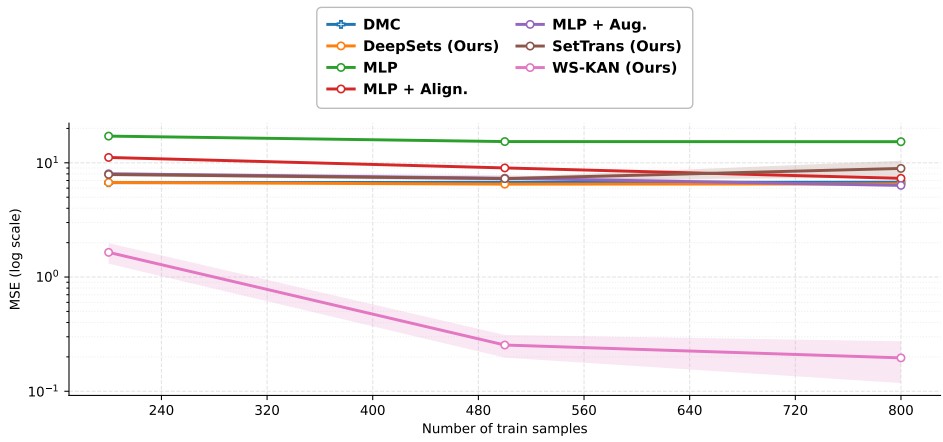

Figure 8: Results on the synthetic 2D sine wave INR dataset.

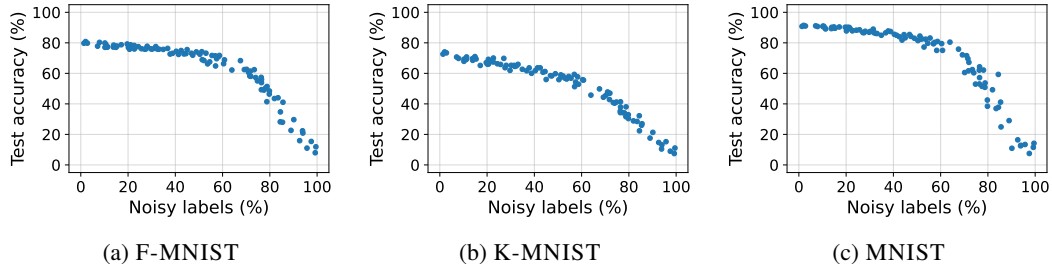

(a) F-MNIST  (b) K-MNIST  (c) MNIST

Figure 9: **KANs test accuracies.** Scatter plots of KAN accuracies on the test set (Y-axis) as a function of the number of noisy labels (X-axis).

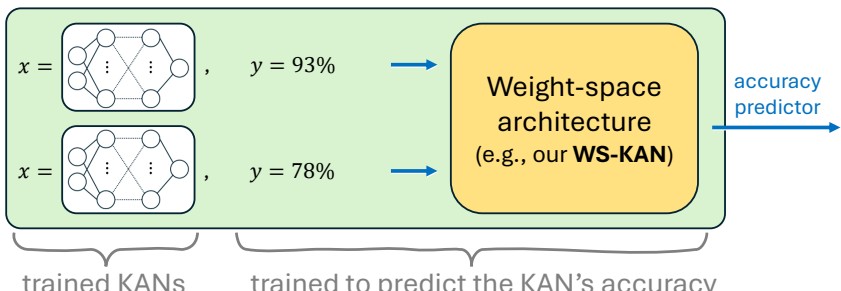

Figure 10: **Weight-space training for accuracy prediction**. We train KANs under varying noise levels, then fit a WS model (e.g., WS-KAN) to predict their accuracy from weights.

validation, and testing. Each KAN model we trained (to classify the images in the dataset at hand) used 3 layers. For MNIST and F-MNIST, the layer dimensions were $[784, 32, 32, 10]$, where $784 = 28 \times 28$ corresponds to the number of grayscale input pixels. For CIFAR10, the dimensions were $[3072, 32, 32, 10]$, with $3072 = 32 \times 32 \times 3$ accounting for the RGB input pixels. Note that 10 denotes the number of target classes across all datasets.

We observed that KANs trained on these datasets achieved similar accuracy levels. To increase the difficulty of predicting accuracy from parameters, we introduced label noise. Specifically, for each KAN, we randomly selected portions of the training data and permuted their labels. In Figure 9, we report the test accuracies over a random sample of 100 test instances.

We trained those KANs for 100 epochs, with a fixed learning rate of 0.01 and a batch size of 128. We again used a grid size of $G = 5$ and $k = 3$ for the B-splines that define the univariate functions in the trained KANs.

### C.2.2 ACCURACY PREDICTION – ADDITIONAL DETAILS

**Hyperparameters.** The hyperparameters used in all experiments, across tasks, datasets, and baselines, are summarized in Table 5.

| Parameter | Values |
|---|---:|
| Embedding size | $\{128, 32\}$ |
| Number of epochs | 100 |
| Learning rate | $\{0.001, 0.005, 0.0001\}$ |
| Dropout | 0.2 |
| Number of layers | $\{4, 1\}$ |
| Batch size | 32 |

Table 5: Hyperparameter configuration used in the experiments for accuracy prediction.

## C.3   PRUNING MASK PREDICTION – EXTENDED SECTION

The pipeline for training a WS model for the task of predicting the pruning mask is provided in Figure 11.

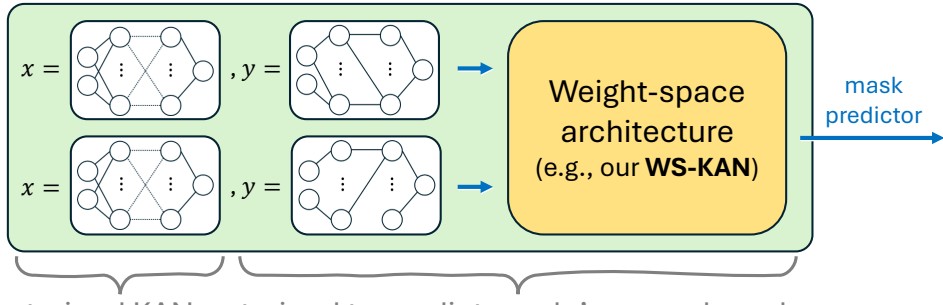

Figure 11: **Pipeline for training a WS model for the task of pruning mask prediction.**

### C.3.1   MODEL ZOO – PRUNING

The model zoos used in these experiments build upon those described in App. C.2.1. More specifically, starting from the KANs obtained in App. C.2.1, we applied the *Oracle prune* algorithm—an edge-pruning method provided by the PyKAN library[7]. Specifically, all edges whose average activation on the training dataset fell below a fixed threshold were removed. Throughout all experiments, we used a threshold of 0.01.

### C.3.2   PRUNING MASK PREDICTION – ADDITIONAL DETAILS AND RESULTS

We note that for the test metrics ROC-AUC and Accuracy reported in Table 4, we used the full set of 500 test samples. Since Accuracy is a threshold-dependent metric, we selected the threshold by maximizing validation accuracy and applied the same threshold during testing.

For the pruning plots for each dataset, as shown in Figures 5a, 5b, 12 and 13, we used a random subset of 100 KANs from the test set, the same as the one for Figure 9.This decision stems from the high computational cost of generating such plots, which requires full inference on the entire dataset (e.g., MNIST) for each pruned KAN produced by every WS method.

**Hyperparameter.**   For all datasets, we adopted a consistent training setup, closely aligned with the configuration described in App. C.1. Specifically, we used a 4-layer architecture with a hidden dimension of 128, a batch size of 32, weight decay of 0.01, and dropout of 0.2. Training was performed for 15 epochs. For the learning rate, we conducted a grid search over $\{0.001, 0.0001\}$.

**Additional results.**   In Figures 12 and 13, we present the downstream pruning results on F-MNIST and K-MNIST, respectively. As shown in Figure 12, WS-KAN achieves the best performance, striking the most favorable trade-off between accuracy and parameter sparsity, closely approaching the oracle prune that serves as our supervision. Meanwhile, the other baselines perform markedly worse. In contrast, Figure 13 shows that DS is also highly effective, performing on par with WS-KAN.

## C.4   ABLATION: POSITIONAL ENCODING AND BIDIRECTIONAL MESSAGE PASSING

Here we present our ablation study to evaluate the importance of both the positional encoding and the bidirectional message passing. Specifically, we report all results in the paper without the positional encoding and the bidirectional message passing.

---

[7]https://github.com/KindXiaoming/pykan.

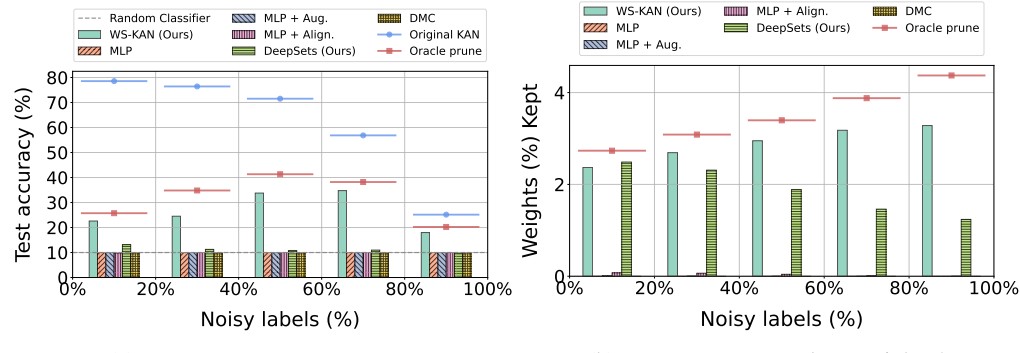

(a) Downstream prune accuracy.     (b) Downstream prune kept weight %.

Figure 12: **Downstream pruning performance across methods over F-MNIST.** We report: (i) **Test accuracy**: the downstream accuracy of pruned networks, averaged over non-overlapping bins of 20%, to highlight the relative effectiveness of pruning strategies under varying noise levels – Figure 12a; (ii) **Kept weights**: the percentage of weights retained after pruning, averaged over the same bins as in (i) – Figure 12b.

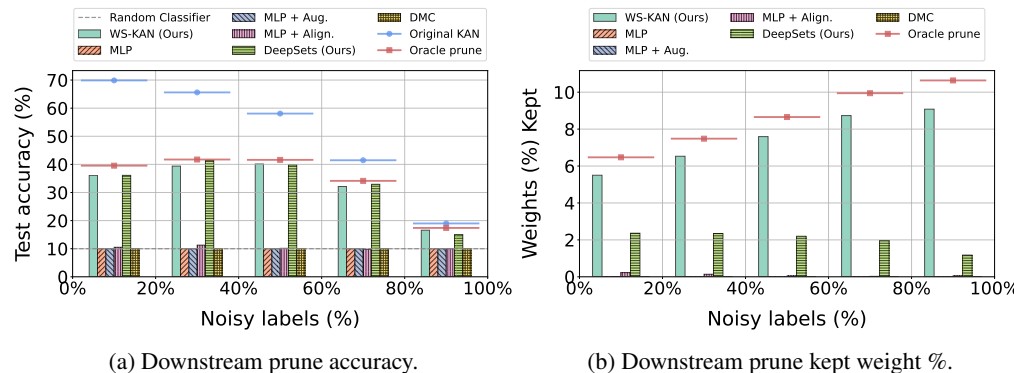

(a) Downstream prune accuracy.     (b) Downstream prune kept weight %.

Figure 13: **Downstream pruning performance across methods over K-MNIST.** We report: (i) **Test accuracy**: the downstream accuracy of pruned networks, averaged over non-overlapping bins of 20%, to highlight the relative effectiveness of pruning strategies under varying noise levels – Figure 13a; (ii) **Kept weights**: the percentage of weights retained after pruning, averaged over the same bins as in (i) – Figure 13b.

**INR Classification Accuracy.** The results are provided in Table 6.

Table 6: **Ablation study** – INR classification accuracy.

| Models | MNIST | F-MNIST | CIFAR10 |
|---|---|---|---|
| **WS-KAN– no PE** | $\mathbf{94.3 \pm 0.4}$ | $84.5 \pm 0.1$ | $\mathbf{46.1 \pm 1.2}$ |
| **WS-KAN– no BIDIR** | $93.8 \pm 0.2$ | $83.4 \pm 0.1$ | $10.0 \pm {<}0.1$ |
| **WS-KAN** | $\mathbf{94.3 \pm 0.5}$ | $\mathbf{84.6 \pm 0.6}$ | $42.2 \pm 0.8$ |

**Accuracy Prediction.** The results are provided in Tables 7 and 8.

Table 7: **Ablation study** – Accuracy prediction (MSE) [$\times 10^3$].

| Model | MNIST (MSE) | F-MNIST (MSE) | K-MNIST (MSE) |
|---|---|---|---|
| **WS-KAN– no PE** | $\mathbf{2.53 \pm 0.22}$ | $\mathbf{2.44 \pm 0.50}$ | $\mathbf{1.10 \pm 0.17}$ |
| **WS-KAN– no BIDIR** | $4.03 \pm 0.62$ | $3.64 \pm 0.12$ | $1.50 \pm 0.08$ |
| **WS-KAN** | $3.29 \pm 0.17$ | $2.94 \pm 0.13$ | $1.45 \pm 0.08$ |

Table 8: **Ablation study** – Accuracy prediction ($R^2$).

| Model | MNIST ($R^2$) | F-MNIST ($R^2$) | K-MNIST ($R^2$) |
|---|---|---|---|
| **WS-KAN– no PE** | $\mathbf{96.00 \pm 0.35}$ | $\mathbf{93.57 \pm 1.32}$ | $\mathbf{96.62 \pm 0.51}$ |
| **WS-KAN– no BIDIR** | $93.64 \pm 0.98$ | $90.40 \pm 0.31$ | $95.54 \pm 0.23$ |
| **WS-KAN** | $94.81 \pm 0.27$ | $92.27 \pm 0.35$ | $95.69 \pm 0.24$ |

**Pruning Mask Prediction.** The results are provided in Tables 9 and 10.

Table 9: **Ablation study** – Pruning mask prediction accuracy.

| Model | MNIST (Accuracy) | F-MNIST (Accuracy) | K-MNIST (Accuracy) |
|---|---|---|---|
| **WS-KAN– no PE** | $97.77 \pm 0.28$ | $98.68 \pm 0.04$ | $96.28 \pm 0.45$ |
| **WS-KAN– no BIDIR** | $96.74 \pm 0.27$ | $98.27 \pm 0.04$ | $95.39 \pm 0.14$ |
| **WS-KAN** | $\mathbf{97.93 \pm 0.19}$ | $\mathbf{98.93 \pm 0.05}$ | $\mathbf{97.72 \pm 0.14}$ |

Table 10: **Ablation study** – Pruning mask prediction AUC.

| Model | MNIST (AUC) | F-MNIST (AUC) | K-MNIST (AUC) |
|---|---|---|---|
| **WS-KAN– no PE** | $99.51 \pm 0.09$ | $99.59 \pm 0.02$ | $98.85 \pm 0.21$ |
| **WS-KAN– no BIDIR** | $98.38 \pm 0.20$ | $98.99 \pm 0.01$ | $97.11 \pm 0.37$ |
| **WS-KAN** | $\mathbf{99.54 \pm 0.01}$ | $\mathbf{99.72 \pm 0.02}$ | $\mathbf{99.46 \pm 0.09}$ |

We observe that WS-KAN performs well even without positional encoding. However, on the more challenging equivariant task—pruning mask prediction—WS-KAN with positional encoding outperforms the version without it in all 6/6 cases. This indicates that positional encoding plays a particularly important role in enhancing WS-KAN's performance on equivariant tasks. We also note that bidirectional message passing is crucial: across all tasks and dataset combinations, WS-KAN without bidirectional message passing consistently underperforms the version that includes it.

### C.5 COMPLEXITY AND RUNTIME ANALYSIS

We report the running times observed in the INR classification tasks. Table 11 summarizes the average training and testing time per epoch (in seconds), computed over 10 epochs and presented together with the corresponding standard deviation. The results indicate that our runtimes are consistent with those commonly observed for standard GNN models on typical graph benchmarks.

**Complexity analysis.** We assume the input KAN contains $n_l$ nodes at layer $l \in \{1, \ldots, L\}$, and that all nodes and edges share the same feature dimensionality, which we treat as a constant. The total number of nodes is therefore

$$N = \sum_{l=1}^{L} n_l,$$

and the total number of edges—due to full connectivity between consecutive layers—is

$$E = \sum_{l=1}^{L-1} n_l \, n_{l+1}.$$

Referring to our message-passing mechanism, as presented in Section 3.3, we obtain the following:

(a) The function $\mathtt{MLP}_v^{(1;F)}$ is applied once per edge, giving $E$ operations. Aggregating over the incoming neighbors of each node contributes another $E$ operations, since the graph is directed and each edge is counted once. Thus, step (a) costs $O(E)$.

(b) This step mirrors (a), except that the edge directions are transposed. Therefore, it also costs $O(E)$.

    (c) This step applies an MLP independently to each edge, contributing another $O(E)$ operations.

    (d) This step is a per-node operation, giving a cost of $O(N)$. Because $E > N$, the edge-wise computations dominate the total cost.

Hence, the overall complexity is

$$O(E).$$

Table 11: Running times measured (in seconds) for all experiments conducted on the INR classification task.

| Dataset | MNIST | F-MNIST | CIFAR-10 |
|---|---|---|---|
| Train time per epoch (s) | $10.12 \pm 0.06$ | $10.18 \pm 0.05$ | $10.88 \pm 0.10$ |
| Test time per epoch (s) | $2.84 \pm 0.12$ | $2.81 \pm 0.06$ | $3.05 \pm 0.08$ |

## D   ALIGNING KOLMOGOROV-ARNOLD NETWORKS

A complementary strategy to designing permutation-equivariant architectures, is to instead keep the model class simple and canonize the input. In our context, the core idea would be to map each KAN into a fixed canonical form by permuting its neurons so that two networks differing only by permutations (and thus computing the same function) are aligned to the same representation.

A particularly interesting case arises in the context of MLPs as discussed in Ainsworth et al. (2022). In that case, $\Theta$'s are collection of scalers, and a simple choice of dist to be the Euclidean distance $\|\cdot\|_2$, the alignment problem becomes

$$\operatorname*{argmin}_{\pi} \ \big\|\operatorname{vec}(\Theta_A) - \operatorname{vec}\big(\pi(\Theta_B)\big)\big\|_2, = \operatorname*{argmax}_{\pi} \ \operatorname{vec}(\Theta_A) \cdot \operatorname{vec}\big(\pi(\Theta_B)\big). \tag{7}$$

We can re-express this in terms of the full weights,

$$\operatorname*{argmax}_{\pi=\{P_i\}} \langle \mathbf{W}_1^{(A)}, \, \boldsymbol{P}_1 \mathbf{W}_1^{(B)} \rangle_F + \langle \mathbf{W}_2^{(A)}, \, \boldsymbol{P}_2 \mathbf{W}_2^{(B)} \boldsymbol{P}_1^\top \rangle_F + \cdots + \langle \mathbf{W}_L^{(A)}, \, \mathbf{W}_L^{(B)} \boldsymbol{P}_{L-1}^\top \rangle_F. \tag{8}$$

Turning to KANs, the parameters $\Theta_A$ and $\Theta_B$ correspond to *collections of univariate functions*, rather than scalars. To align such models, we must therefore define a distance function appropriate for functions. Importantly, there is no single "correct" choice here—different definitions may lead to different alignment outcomes.

In this paper, consistent with our formalism in Section 3.2, we associate each such entry in $\Theta$, with a parameter vector of the form

$$(\Theta_A)_i = [w_b, \ w_s, \ \mathbf{c}],$$

where, for the sake of this example and with a slight abuse of notation, $w_b$, $w_s$, and $\mathbf{c}$ are the coefficients defining the corresponding $i$-th 1D function. For convenience, we refer to these coefficients as *channels*, and denote the $c$-th channel of this vector by $(\Theta_A^c)_i = [w_b, \ w_s, \ \mathbf{c}]_c$.

With this structure, we define the alignment objective as the sum of $\ell_2$ distances across all channels:

$$\arg\min_{\pi} \ \sum_c \ \big\|\operatorname{vec}(\Theta_A^c) - \operatorname{vec}\big(\pi(\Theta_B^c)\big)\big\|_2 = \arg\max_{\pi} \ \sum_c \ \operatorname{vec}(\Theta_A^c) \cdot \operatorname{vec}\big(\pi(\Theta_B^c)\big), \tag{9}$$

which parallels Eq. (7), but now summed across channels. Expanding this channel-wise objective yields

$$\operatorname*{argmax}_{\pi=\{P_i\}} \sum_c \left( \langle \phi_1^{(c;A)}, \, \boldsymbol{P}_1 \phi_1^{(c;B)} \rangle_F + \langle \phi_2^{(c;A)}, \, \boldsymbol{P}_2 \phi_2^{(c;B)} \boldsymbol{P}_1^\top \rangle_F + \cdots + \langle \phi_L^{(c;A)}, \, \phi_L^{(c;B)} \boldsymbol{P}_{L-1}^\top \rangle_F \right)$$

$$= \left( \operatorname*{argmax}_{\pi=\{P_i\}} \langle \phi_1^{(A)}, \, \boldsymbol{P}_1 * \phi_1^{(B)} \rangle_F + \langle \phi_2^{(A)}, \, \boldsymbol{P}_2 * \phi_2^{(B)} * \boldsymbol{P}_1^\top \rangle_F + \cdots + \langle \phi_L^{(A)}, \, \phi_L^{(B)} * \boldsymbol{P}_{L-1}^\top \rangle_F \right).$$

In the second line, we move the sum over channels $c$ inside each Frobenius inner product. As a result, the channel superscript $^c$ is omitted: the Frobenius inner product now implicitly sums over $c$, absorbing the channel dimension. Importantly, $*$ denotes a contraction operator that acts only on the first two indices, leaving the channel dimension untouched. Concretely, for the middle term we have:

$$(\boldsymbol{P}_2 * \phi_2^{(B)} * \boldsymbol{P}_1^\top)_{p,q,c} := \sum_{p',q'} (\boldsymbol{P}_2)_{p,p'} \, (\phi_2^{(B)})_{p',q',c} \, (\boldsymbol{P}_1^\top)_{q,q'}. \tag{10}$$

This formalism provides a straightforward extension of the alignment framework of Ainsworth et al. (2022) for MLPs to the KAN setting: their algorithm can be directly applied once the $*$ operation is defined as above.

### D.1 Aligning a dataset of trained KANs

While the formalism in App. D provides a method for aligning a pair of KANs, our experiments require aligning an entire dataset of KANs. To achieve this, we integrate our alignment technique with the MergeMany algorithm introduced in Ainsworth et al. (2022). The central idea is to iteratively run the alignment procedure between one model and the average of all other models.

In practice, we observed that convergence times could be prohibitively long. To address this, we adopted the following strategy: we fixed the error tolerance at 0.001, then randomly sampled 1000 models and applied the MergeMany algorithm to this subset, consistent with Navon et al. (2023). The outcome was a new model representing the average of the aligned subset. This model then served as the reference for aligning the remaining training models as well as the test models, ultimately yielding fully aligned training and test sets.

Finally, we note that in our formulation, all channels are equally weighted. While this simplifies the alignment, it may be suboptimal in certain cases. Alternative weighting schemes could yield improved results. We leave a detailed exploration of such extensions to future work.

## E  Proofs

**Lemma E.1** (MLP as an approximation of the univariate functions composing the KAN). *Let $\mathcal{B}$ denote the family of cardinal B-splines of degree $k$, defined on a fixed grid $G$ over the domain $[a,b]$, and parameterized by coefficients $\boldsymbol{c} = [c_0, \ldots, c_{G+k-1}] \in \mathcal{C} \subset \mathbb{R}^{G+k}$, where the set of admissible coefficients lies in a compact domain $\mathcal{C}$. Consider the function $\psi : \mathbb{R} \to \mathbb{R}$ defined as $\psi(x) = w_b \, b(x) + w_s \, B(x)$, where $w_b, w_s \in \mathcal{W} \subset \mathbb{R}$ for some compact set $\mathcal{W}$. $b(x)$ denotes the silo function, and $B(x)$ denotes the B-spline. Then, for any $\varepsilon > 0$, and for any compact domain $\mathcal{X}$ there exists a set of weights for a multilayer perceptron $\mathtt{MLP} : \mathbb{R}^{G+k+3} \to \mathbb{R}$ such that,*

$$\sup_{x \in \mathcal{X}} \big| \mathtt{MLP}(x, w_s, w_b, \boldsymbol{c}) - \psi(x) \big| < \varepsilon.$$

*Proof.* The claim follows directly from the universal approximation theorem Cybenko (1989). The input vector to the $\mathtt{MLP}$ lies in the compact domain

$$(x, w_s, w_b, \boldsymbol{c}) \in \mathcal{X} \times \mathcal{W} \times \mathcal{W} \times \mathcal{C},$$

which is compact since finite cartesian products of compact sets are compact. Moreover, the target function $\psi(\cdot)$ is continuous, as it is a weighted sum of continuous functions—both the silo function $b(\cdot)$ and the B-spline $B(\cdot)$ are continuous. Therefore, by the universal approximation theorem, there exists an $\mathtt{MLP}$ that uniformly approximates $\psi$ on this compact domain to arbitrary accuracy $\varepsilon > 0$. This completes the proof. $\qquad\square$

**Proposition E.2** (WS-KAN can simulate the forward pass of KANs). *Let $f_\theta$ be a given KAN architecture, defined over an input domain $[a,b]^n$, where each univariate function is represented by a B-spline from the family $\mathcal{B}$. Let $G$ be its KAN-graph, where the nodes in the first layer are enhanced with the input $\boldsymbol{x} \in [a,b]^n$. For every $\varepsilon > 0$, there exists a WS-KAN such that,*

$$\sup_{\boldsymbol{x} \in [a,b]^n} \big| \mathit{WS\text{-}KAN}(G) - f_\theta(\boldsymbol{x}) \big| < \varepsilon.$$

*Proof.* We recall that $f_\theta(\boldsymbol{x}) = (\boldsymbol{\phi}^L \circ \ldots \circ \boldsymbol{\phi}^1)\boldsymbol{x}$ is a composition of continuous functions. We denote by $\boldsymbol{v}^l$ the nodes in the KAN-graph that correspond to the activations in layer $l$, defined $\mathbf{a}^l$. We prove via induction, that $l+1$ layers of WS-KAN, can uniformly approximate the output $\mathbf{a}^{l+1} = (\boldsymbol{\phi}^{l+1} \circ \ldots \circ \boldsymbol{\phi}^1)\boldsymbol{x}$ over the nodes $\boldsymbol{v}^{l+1}$ to arbitrary precision.

**Induction proof.** We begin by defining a compact space, $\mathcal{S}$, which encompasses all possible values attained by any neuron within the KAN. Since the input to the KAN lies in $[a,b]^n$ (which is compact) and the applied univariate functions are continuous, their outputs form compact sets. With a mild abuse of terminology, we define the compact output space to be a compact set that includes the outputs corresponding to all inputs in the unit ball $B(x,1)$ around each input. Each neuron then receives a finite sum of elements, each belonging to a compact set, which implies that the neuron's value also lies within a compact set. Moreover, because the number of neurons is finite, the union of all such compact sets is itself contained in a larger compact domain. Thus, there exists a compact space, defined as $\mathcal{S}$, that contains the values of all neurons. Additionally, we denote the maximal width of the network by $N_{\max}$.

We now proceed with a proof by induction.

**Step 1. Initialization.** We define the KAN's input over the *KAN-graph* as follows,

$$\boldsymbol{v}^0 = \boldsymbol{x}, \quad \boldsymbol{v}^{l>0} = \boldsymbol{0}, \quad \boldsymbol{e}^l_{p,q} = \tilde{\boldsymbol{\phi}}^l_{p,q} := [w^l_{b;p,q}, \, w^l_{s;p,q}, \, \boldsymbol{c}^l_{p,q}],$$

so that the node features at layer $0$ encode the KAN's input $\boldsymbol{x}$, while edge features store the spline parameters. Thus, the base case is satisfied.

**Step 2. Induction hypothesis.** We assume that after $l$ layers of WS-KAN message passing we have

$$\|\boldsymbol{v}^l - \boldsymbol{a}^l\| < \delta,$$

where $\boldsymbol{a}^l$ denotes the activations at layer $l$ of the KAN, and $\delta > 0$ is an arbitrarily small approximation error **that we can control**. Given $\epsilon > 0$, we now show that we are able to obtain $\boldsymbol{v}^{l+1}$ such that $\|\boldsymbol{v}^{l+1} - \boldsymbol{a}^{l+1}\| < \epsilon$.

**Step 3. Inductive step.**

We recall a given WS-KAN update equation:

$$\boldsymbol{v}^{\mathrm{F}}_i \leftarrow \mathtt{MLP}^{(2;\,\mathrm{F})}_v\Big(\boldsymbol{v}_i, \sum_{j:\,\boldsymbol{e}(i,j)\in E} \mathtt{MLP}^{(1;\,\mathrm{F})}_v(\boldsymbol{v}_j, \boldsymbol{e}_{(i,j)})\Big), \quad \boldsymbol{v}^{\mathrm{B}}_i \leftarrow \mathtt{MLP}^{(2;\,\mathrm{B})}_v\Big(\boldsymbol{v}_i, \sum_{j:\,\boldsymbol{e}(i,j)\in E^T} \mathtt{MLP}^{(1;\,\mathrm{B})}_v(\boldsymbol{v}_j, \boldsymbol{e}_{(i,j)})\Big),$$

$$\boldsymbol{e}_{(i,j)} \leftarrow \mathtt{MLP}_e(\boldsymbol{v}_i, \boldsymbol{v}_j, \boldsymbol{e}_{(i,j)}), \qquad\qquad \boldsymbol{v}_i \leftarrow \mathtt{MLP}^{(3)}_v(\boldsymbol{v}_i, \boldsymbol{v}^{\mathrm{F}}_i, \boldsymbol{v}^{\mathrm{B}}_i).$$

The KAN update rule at layer $l+1$ is

$$\boldsymbol{a}^{l+1}_p = \sum_q \boldsymbol{\phi}^l_{p,q}(\boldsymbol{a}^l_q),$$

where $\boldsymbol{\phi}^l_{p,q}$ denotes a univariate spline function parameterized by $(w^l_{b;p,q}, w^l_{s;p,q}, \boldsymbol{c}^l_{p,q})$.

To reproduce this update within WS-KAN, we recall that $\mathtt{MLP}^{\text{univariate}}$ can be chosen to be an arbitrarly close approximation of the univariate function composting the KAN, recall Lemma 4.1, thus we choose $\mathtt{MLP}^{\text{univariate}}$, such that

$$\|\mathtt{MLP}^{\text{univariate}}(a, b, c, d) - \psi(a)\| < \epsilon/2N_{\max},$$

where $\psi$ stands for the 1D univariate functions with the parameters $(b, c, d)$.

Since $\mathtt{MLP}^{\text{univariate}}(a, b, c, d)$ is continuous and defined on a compact set, it is uniformly continuous. Thus, there exists $\delta' > 0$ such that,

$$\|a - a'\| < \delta' \Rightarrow \|\mathtt{MLP}^{\text{univariate}}(a, b, c, d) - \mathtt{MLP}^{\text{univariate}}(a', b, c, d)\| < \epsilon/2N_{\max}.$$

Thus, we fix $\delta < \min(\delta', 1)$, and from the induction hypothesis we get that there is a WS-KAN network that can guarantee $\|\boldsymbol{v}^l - \boldsymbol{a}^l\| < \delta$.

It holds that for a single neruon, given that $\|a - a'\| < \delta$,

$$\left\|\text{MLP}^{\text{univariate}}(a', b, c, d) - \psi(a)\right\| =$$
$$\left\|\text{MLP}^{\text{univariate}}(a', b, c, d) - \text{MLP}^{\text{univariate}}(a, b, c, d) + \text{MLP}^{\text{univariate}}(a, b, c, d) - \psi(a)\right\| \leq$$
$$\left\|\text{MLP}^{\text{univariate}}(a', b, c, d) - \text{MLP}^{\text{univariate}}(a, b, c, d)\right\| + \left\|\text{MLP}^{\text{univariate}}(a, b, c, d) - \psi(a)\right\| \leq$$
$$\epsilon/2N_{\max} + \epsilon/2N_{\max} = \epsilon/N_{\max}$$

We set,

$$\text{MLP}_v^{(1;F)}\left(\boldsymbol{v}_j, \boldsymbol{e}_{(i,j)}\right) = \text{MLP}^{\text{univariate}}\left(\boldsymbol{v}_q^l, w_{b;p,q}^l, w_{s;p,q}^l, \boldsymbol{c}_{p,q}^l\right). \tag{11}$$

Thus, it holds that for $\|\boldsymbol{v}_q^l - \boldsymbol{a}_q^l\| < \delta$, which is our induction assumption, we have,

$$\|\text{MLP}_v^{(1;F)}\left(\boldsymbol{v}_j, \boldsymbol{e}_{(i,j)}\right) - \phi_{p,q}^l(\boldsymbol{a}_q^l)\| < \epsilon/N_{\max},$$

We can also assign weight to the following MLP, to output the following precisely,

$$\text{MLP}_v^{(2;F)}(a, b) = b, \tag{12}$$
$$\tag{13}$$

The forward aggregator $\boldsymbol{v}_i^{\text{forward}}$ thus becomes

$$|v_q^{\text{forward}} - \sum_q \phi_{p,q}^l(a_q^l)| \leq \epsilon/N_{\max},$$

which approximates the KAN update in this layer.

The backward aggregator, edge update, and auxiliary terms are irrelevant for this construction, so their associated MLPs, namely are set to the zero function, by assigning zero to all weight:

$$\text{MLP}_v^{(1;B)} = \text{MLP}_v^{(2;B)} = \text{MLP}_e = 0, \tag{14}$$

Finally, we set the weights of $\text{MLP}_v^{(3)}$ such that $\text{MLP}_v^{(3)}(a, b, c) = b$. this ensures that $\boldsymbol{v}_q^{l+1} = \boldsymbol{v}_q^{\text{forward}}$. Thus.

$$\|\boldsymbol{v}^{l+1} - \boldsymbol{a}^{l+1}\|_2 = \left(\sum_p \left(v_p^{l+1} - a_p^{l+1}\right)^2\right)^{\frac{1}{2}} \leq (\sum_p (\epsilon/N_{\max})^2)^{\frac{1}{2}} \leq \epsilon/\sqrt{N_{\max}} < \epsilon$$

$\square$

**Proposition E.3** (KAN symmetries). *Let $\theta = (\phi^L, \ldots, \phi^1)$ denote the collection of parametric one-dimensional functions composing an L-layer KAN. Consider the group, $G \coloneqq S_{d_1} \times S_{d_2} \times \cdots \times S_{d_{L-1}}$, the direct product of symmetric groups corresponding to the intermediate dimensions $d_1, \ldots, d_{L-1}$. Let $g = (\boldsymbol{P}_1, \ldots, \boldsymbol{P}_{L-1}) \in G$, where each $\boldsymbol{P}_l$ is the permutation matrix of $\sigma_l \in S_{d_l}$. Define the group action $g \cdot \theta = \theta'$ with $\theta' = (\phi'^L, \ldots, \phi'^1)$ given by,*

$$\phi'^1 = \boldsymbol{P}_1^\top \phi^1, \qquad \phi'^l = \boldsymbol{P}_L^\top \phi^l \boldsymbol{P}_{l-1}, \ \forall l = 2, \ldots, L-1, \qquad \phi'^L = \phi^L \boldsymbol{P}_{L-1}.$$

*Then, $f_\theta(\boldsymbol{x}) = f_{\theta'}(\boldsymbol{x})$ for all $\boldsymbol{x}$.*

*Proof.* We compute the $p$-th output component:

$$f_\theta(\boldsymbol{x})_p = (\phi^2 \circ \phi^1 \circ \boldsymbol{x})_p = \sum_{q,k} \phi_{p,q}^2(\phi_{q,k}^1(\boldsymbol{x}_k)) = \sum_{q,k} \phi_{p,\sigma(q)}^2(\phi_{\sigma(q),k}^1(\boldsymbol{x}_k))$$
$$= \sum_{q,k} (\phi^2 \boldsymbol{P})_{p,q}(\boldsymbol{P}^\top \phi^1)_{q,k}(\boldsymbol{x}_k) = f_{\tilde{\theta}}(\boldsymbol{x})_p$$

The third equality holds by reindexing the summation using a permutation $\sigma$ corresponding to $\boldsymbol{P}$. Recalling Eq. (4), the fourth equality follows from the identities $(\phi^2 \boldsymbol{P})_{p,q} = \phi_{p,\sigma(q)}^2$ and $(\boldsymbol{P}^\top \phi^1)_{q,k} = \phi_{\sigma(q),k}^1$. $\square$

## F    LARGE LANGUAGE MODEL (LLM) USAGE

We used large language models (LLMs) exclusively to support the writing process. Their role was limited to improving clarity in technical explanations, refining grammar and style, and enhancing overall readability. All research contributions—including experimental design, data analysis, and conclusions—are entirely our own. The LLMs served solely as tools to improve presentation quality and were not employed to generate research content or influence the substance of our work.

