# OpenReview forum: "A Graph Meta-Network for Learning on Kolmogorov–Arnold Networks"
_ICLR.cc/2026/Conference — ICLR 2026 Poster_

### Official Review · Reviewer_vi4a · 2025-10-27

**Soundness:** 3
**Presentation:** 3
**Contribution:** 3
**Rating:** 8
**Confidence:** 3

**Summary:**

The paper introduces a graph-based framework for designing meta-networks that operate on the pretrained weights of Kolmogorov–Arnold Networks (KANs).

**Strengths:**

The paper is well-written and self-contained. Although I am not very familiar with the recent developments on KANs, I was able to understand the architecture clearly. The authors provide a good demonstration of the symmetry properties of the KAN parameter space, along with a clear explanation of how a KAN can be interpreted as a graph. Their approach closely follows the standard methodology used in the literature for constructing metanetworks for traditional MLPs and CNNs.

**Weaknesses:**

I did not identify any clear weaknesses. Only a few minor typos were found, such as “silo function” in Lemma 4.1 or “weight-space model” in line 10.

**Questions:**

I have a question regarding the permutation symmetry of KANs presented in Section 3.1. While it is clear that the permutation symmetry preserves the network’s functional behavior, I am wondering whether there exist other types of symmetry. For example, in MLPs and CNNs with ReLU activations, there is a well-known scaling symmetry, which has also been leveraged in the design of graph-based metanetworks [1]. A similar analysis for KANs would help clarify whether additional symmetries exist beyond permutation, and would further justify that modeling permutation symmetry alone is sufficient for constructing the proposed graph-based network.


[1] Ioannis Kalogeropoulos, Giorgos Bouritsas, and Yannis Panagakis. Scale equivariant graph metanetworks.

---

> ### Author Response · Authors · 2025-11-21
> **Response to reviewer vi4a**
>
> We thank the reviewer for the positive feedback. We are pleased that the reviewer has appreciated our demonstration of the symmetry properties of KANs parameter space, and our KAN-graph construction. Below we address the reviewer's questions and comments.
>
> ---
>
> ***"I did not identify any clear weaknesses. Only a few minor typos were found, such as “silo function” in Lemma 4.1 or “weight-space model” in line 10."***
>
> We thank the reviewer for his attention to detail, we will fix the typos in the revised version.
>
> ---
>
> ***"I have a question regarding the permutation symmetry of KANs presented in Section 3.1. While it is clear that the permutation symmetry preserves the network’s functional behavior, I am wondering whether there exist other types of symmetry. For example, in MLPs and CNNs with ReLU activations, there is a well-known scaling symmetry, which has also been leveraged in the design of graph-based metanetworks [1]. A similar analysis for KANs would help clarify whether additional symmetries exist beyond permutation, and would further justify that modeling permutation symmetry alone is sufficient for constructing the proposed graph-based network."***
>
> We thank the reviewer for bringing up this interesting point. We haven't found any other symmetries that extend to KANs. For example, the scaling symmetries that are present in MLPs and CNNs, are not present in the standard KANs, as its 1D functions are typically based on B-splines, which are not invariant to scaling. We believe this is an interesting question for future work -- as KANs continue to evolve, their 1D functions will likely change and introduce more symmetries, which will be interesting to explore.

---

> > ### Comment · Reviewer_vi4a · 2025-11-24
> >
> > I thank the Authors for their response!
> >
> > My main question in the review concerns the symmetry of KANs. While it is clear that common symmetries, such as scaling, do not directly apply to KANs, it would be helpful to clarify whether any additional forms of symmetry exist or, if not, to provide a theoretical justification. The Authors may consider adding a brief discussion on this point in the manuscript for greater clarity.
> >
> > Nonetheless, I understand that the primary focus of the paper is the construction of a graph-based metanetwork for KANs, with an emphasis on permutation symmetry. With this in mind, I am happy to maintain my positive score.

---

### Official Review · Reviewer_y2se · 2025-10-28

**Soundness:** 2
**Presentation:** 3
**Contribution:** 2
**Rating:** 4
**Confidence:** 3

**Summary:**

This paper studies “learning in weight space” for Kolmogorov–Arnold Networks (KANs). The authors (i) prove KANs admit the same hidden-neuron permutation symmetries as MLPs, (ii) encode a KAN as a directed “KAN-graph” whose edge features are the parameters of each learned univariate function, and (iii) process that graph with a GNN they call WS-KAN. They further show WS-KAN can simulate the forward pass of an input KAN (a standard expressivity criterion in weight-space works), then build several “model zoos” of trained KANs and evaluate on INR classification, accuracy prediction, and pruning-mask prediction—reporting consistent gains over baselines and large speedups vs an oracle pruning routine that requires data passes.

**Strengths:**

1. Concrete construction of edge features for KANs’ 1D functions; positional encodings are explained and motivated.

2. A standard expressivity result: WS-KAN can approximate a KAN’s forward pass, with proof sketch and full appendix.

**Weaknesses:**

1. **Lack of empirical symmetry validation:** The work does not experimentally verify the equivariance or invariance properties that the theory implies.

2. **Limited range of equivariant tasks:** Prior works such as Editing INRs [1] and Learning to Optimize [2] evaluate equivariant models on richer, more practical tasks. Incorporating similar benchmarks would strengthen the empirical relevance.

[1] Zhou et al., Permutation Equivariant Neural Functionals. NeurIPS 2023.

[2] Kofinas et al., Graph Neural Networks for Learning Equivariant Representations of Neural Networks. ICLR 2024.

3. **Unaddressed scalability issues:** The paper omits discussion of computational complexity and runtime behavior. Since the KAN-graph grows quadratically with network connectivity, an analysis of how WS-KAN scales with KAN depth and width is needed.


4. **Incomplete design analysis:** Key architectural choices—such as the use of bidirectional message passing and specific positional encodings-are insufficiently justified. Ablation studies would clarify their impact on performance

**Questions:**

See weaknesses.

---

> ### Author Response · Authors · 2025-11-21
> **Response to reviewer y2se [1/3]**
>
> We thank the reviewer for the constructive feedback. We are pleased that they appreciate our KAN-graph construction, as well as our theoretical results regarding its expressivity. Below, we address the reviewer's questions and comments.
>
> ---
>
> ***"Lack of empirical symmetry validation: The work does not experimentally verify the equivariance or invariance properties that the theory implies."***
>
> We thank the reviewer for raising this point. We will present an even stronger claim: *any permutation invariant GNN architecture (of which our model is a special case) is guaranteed to produce exactly the same output for any two input KANs that differ only by a permutation of their intermediate nodes.* Below we provide a brief proof sketch.
>
> Let $G_1$ and $G_2$ be two KANs such that $G_2$ is a permuted version of $G_1$, given by a permutation matrix $P$. We emphasize that $P$ corresponds to the permutation matrix of the group element $g$ in Proposition 3.1, once the $\phi$'s composing the KAN are arranged into an adjacency matrix following the inset illustration in lines 202–207.
>
> Concretely, if $G_1 = (X, A)$, then $G_2 = (P X, P A P^{T})$. Our goal is to show that for any GNN architecture $\mathrm{GNN}$, it holds that
>
> $$
> \mathrm{GNN}(P X + \tilde{X},\, P A P^{T} + \tilde{A}) = \mathrm{GNN}(X + \tilde{X},\, A + \tilde{A}),
> $$
>
> where $\tilde{X}, \tilde{A}$ corresponds to our **fixed** positional encodings, which by construction satisfy
>
> $$
> \tilde{X} = P \tilde{X}, \qquad \tilde{A} = P \tilde{A} P^{T}.
> $$
>
> Thus, we obtain
>
> $$
> \mathrm{GNN}(P X + \tilde{X},\, P A P^{T} + \tilde{A}) = \mathrm{GNN}(P X + P \tilde{X},\, P A P^{T} + P \tilde{A} P^{T}) = \mathrm{GNN}\big(P (X + \tilde{X}),\, P (A + \tilde{A}) P^{T}\big) = \mathrm{GNN}(X + \tilde{X},\, A + \tilde{A}),
> $$
>
> where the first equality follows directly from the construction of the positional encodings, and the final equality follows from the fact that any GNN is equivariant to node permutations.
>
>  ---
>
> ***"Limited range of equivariant tasks: Prior works such as Editing INRs [1] and Learning to Optimize [2] evaluate equivariant models on richer, more practical tasks. Incorporating similar benchmarks would strengthen the empirical relevance."***
>
> Respectfully, we disagree with the reviewer’s statement. As the first work to propose and evaluate a weight-space model for KANs, we believe our model zoo provides a comprehensive and domain-appropriate empirical evaluation. It spans both invariant tasks -- INR classification and accuracy prediction -- as well as a challenging equivariant task uniquely suited to KANs, namely, pruning-mask prediction over the edges, which correspond to the 1D functions composing the KANs.
>
> To further support this point, we compare our experimental design to that of the analogous weight-space paper for standard MLPs [1] – the first to introduce an equivariant weight-space architecture for MLPs and, to construct its own model zoo for evaluation. By this standard, our benchmarks and model zoo are at least as rich and, in several respects, more extensive:
>
> 1. **Invariant tasks:**
>
> - Prior work evaluates INR classification on three datasets (a synthetic dataset, MNIST, and Fashion-MNIST).
>
> - We evaluate on four datasets (synthetic, MNIST, Fashion-MNNST, and K-MNIST).
>
> 2. **Equivariant tasks:**
>
> - Prior work considers a single equivariant task on one dataset—domain adaptation on CIFAR-10.
>
> - In contrast, we evaluate pruning-mask prediction, an equivariant task *intrinsic to KANs*, across **three** datasets (MNIST, Fashion-MNIST, and K-MNIST).
>
> 3. **Additional invariant task:**
>
> - Beyond the above, we also include accuracy prediction on the same three datasets, further broadening the empirical coverage.
>
> Taken together, these components constitute a diverse and meaningful suite of benchmarks tailored specifically to the KAN setting, and we believe that they sufficiently demonstrate the empirical relevance and robustness of WS-KAN.
>
> ---
>
> ***References:***
>
>  [1] Equivariant Architectures for Learning in Deep Weight Spaces. ICML 2024.

---

> > ### Author Response · Authors · 2025-11-21
> > **Response to reviewer y2se [2/3]**
> >
> > ***"Unaddressed scalability issues: The paper omits discussion of computational complexity and runtime behavior. Since the KAN-graph grows quadratically with network connectivity, an analysis of how WS-KAN scales with KAN depth and width is needed."***
> >
> > We thanks the reviewer for the opportunity to clarify, as a response to this reviewer's comment, we took the following steps:
> >
> > 1. We computed the running time of all experiments regarding the INR classification task.
> >
> > 2. We provide a complexity analysis of our proposed WS-KAN architecture.
> >
> > Below, we provide more details for each of the steps above.
> >
> > 1. We note that the running time we observed in the INR classification tasks is the standard run-time observed when using standard Graph Neural Networks (GNNs) on standard graph benchmarks. We report the running time (in seconds) per epoch for both training and testing in the table below, averaging over 10 epochs with standard deviation.
> >
> > | Dataset                | MNIST          | F-MNIST        | CIFAR-10       |
> > |------------------------|----------------|----------------|----------------|
> >  | Train time per epoch (s) | 10.12 ± 0.06   | 10.18 ± 0.05   | 10.88 ± 0.10   |
> >  | Test time per epoch (s)  | 2.84 ± 0.12    | 2.81 ± 0.06    | 3.05 ± 0.08    |
> >
> > We observe that the run-time is contained -- around 10 seconds per epoch for training and 3 seconds per epoch for testing.
> >
> > 2. Below we provide a complexity analysis of our proposed WS-KAN architecture.
> > We assume the input KAN contains $n_l$ nodes at layer $l \in \{1, \ldots, L\}$, and that all nodes and edges share the same feature dimensionality and it is a constant. The total number of nodes is therefore:
> >
> > $$
> > N = \sum_{l=1}^L n_l,
> > $$
> >  and the total number of edges -- due to full connectivity between consecutive layers -- is:
> >
> > $$
> > E = \sum_{l=1}^{L-1} n_l \, n_{l+1}.
> > $$
> >
> > Referring to Equations (a–d) in the paper (lines 226–229), we obtain the following:
> >
> >  - **(a)** The function $\texttt{MLP}^{(1;F)}_v$ is applied once per edge, giving $E$ operations. Aggregating over the incoming neighbors of each node contributes another $E$ operations, since the graph is directed and each edge is counted once. Thus, step (a) costs $O(E)$.
> >
> > - **(b)** This step mirrors (a), except that the edge directions are transposed. Therefore, it also costs $O(E)$.
> >
> > - **(c)** This step applies an MLP independently to each edge, contributing another $O(E)$ operations.
> >
> > - **(d)** This step is a per-node operation, giving a cost of $O(N)$. Because $E > N$, the edge-wise computations dominate the total cost.
> >
> > Hence, the overall complexity of a single WS-KAN layer is $ O(E) $ .
> >
> > We have added this to the edited version of the paper, in blue, to Appendix C.6.

---

> > > ### Author Response · Authors · 2025-11-21
> > > **Response to reviewer y2se [3/3]**
> > >
> > > ***"Incomplete design analysis: Key architectural choices—such as the use of bidirectional message passing and specific positional encodings-are insufficiently justified. Ablation studies would clarify their impact on performance"***
> > >
> > > We thank the reviewer for bringing this up, and we conducted an ablation study to evaluate the importance of both the positional encoding and the bidirectional message passing. Specifically, we have reran all experiments in the paper without the positional encoding and the bidirectional message passing, and we report the results in the tables below.
> > >
> > > 1. INR Classification Accuracy
> > >
> > > | Model               | MNIST         | F-MNIST       | CIFAR-10     |
> > > |---------------------|---------------|---------------|--------------|
> > >  | **WS-KAN – no PE**  | **94.3 ± 0.4** | 84.5 ± 0.1 | **46.1 ± 1.2** |
> > > | **WS-KAN – no BIDIR**  | 93.8 ± 0.2 | 83.4 ± 0.1 | 10.0 ± < 0.1 |
> > >  | **WS-KAN**          | **94.3 ± 0.5**    | **84.6 ± 0.6**    | 42.2 ± 0.8    |
> > >
> > >
> > > 2. Accuracy prediction
> > >
> > > | Model | MNIST  (MSE)             | F-MNIST (MSE)             | K-MNIST (MSE)             |
> > >  |-----------|----------------------|----------------------|----------------------|
> > >  | **WS-KAN – no PE**  | **2.53 ± 0.22**    | **2.44 ± 0.50**    | **1.1 ± 0.17**    |
> > > | **WS-KAN – no BIDIR**  | 4.03 ± 0.62    | 3.64 ± 0.12    | 1.5 ± 0.08    |
> > > | **WS-KAN**   | 3.29 ± 0.17          | 2.94 ± 0.13          | 1.45 ± 0.08          |
> > >
> > >
> > >  | Model | MNIST  (R²)             | F-MNIST (R²)             | K-MNIST (R²)             |
> > > |-----------|----------------------|----------------------|----------------------|
> > >  | **WS-KAN – no PE**  | **96.00 ± 0.35**    | **93.57 ± 1.32**    | **96.62 ± 0.51**    |
> > >  | **WS-KAN – no BIDIR**  | 93.64 ± 0.98    | 90.40 ± 0.31    | 95.54 ± 0.23    |
> > > | **WS-KAN**| 94.81 ± 0.27         | 92.27 ± 0.35         | 95.69 ± 0.24         |
> > >
> > > 3. Pruning Mask Prediction
> > >
> > > | Model | MNIST  (Accuracy)             | F-MNIST (Accuracy)             | K-MNIST (Accuracy)             |
> > >  |-----------|----------------------|----------------------|----------------------|
> > > | **WS-KAN – no PE** | 97.77 ± 0.28    | 98.68 ± 0.04    | 96.28 ± 0.45    |
> > >  | **WS-KAN – no BIDIR** | 96.74 ± 0.27    | 98.27 ± 0.04    | 95.39 ± 0.14    |
> > > | **WS-KAN**| **97.93 ± 0.19**         | **98.93 ± 0.05**         | **97.72 ± 0.14**         |
> > >
> > > | Model | MNIST  (AUC)             | F-MNIST (AUC)             | K-MNIST (AUC)             |
> > > |-----------|----------------------|----------------------|----------------------|
> > > | **WS-KAN – no PE** | 99.51 ± 0.09    | 99.59 ± 0.02    | 98.85 ± 0.21    |
> > >  | **WS-KAN – no BIDIR** | 98.38 ± 0.20    | 98.99 ± 0.01    | 97.11 ± 0.37    |
> > > | **WS-KAN**| **99.54 ± 0.01**         | **99.72 ± 0.02**         | **99.46 ± 0.09**         |
> > >
> > > We observe that WS-KAN performs well even without positional encoding. However, on the more challenging equivariant task -- pruning mask prediction -- WS-KAN with positional encoding outperforms the version without it in all 6 out of 6 cases. This indicates that positional encoding plays a more important role in enhancing WS-KAN’s performance on equivariant tasks. We also note that bidirectional message passing is crucial: across all tasks and dataset combinations, WS-KAN without bidirectional message passing consistently underperforms the version that includes it.
> > >
> > > We have added this to the edited version of the paper, in blue, to Appendix C.5.

---

> > > > ### Comment · Reviewer_y2se · 2025-11-24
> > > >
> > > > I thank the authors for the detailed additional results and the revisions to the manuscript. I appreciate the clear explanations provided in the rebuttal, and I am satisfied that all of my main concerns have been fully addressed. I will increase my score to 6.

---

### Official Review · Reviewer_1aub · 2025-10-30

**Soundness:** 2
**Presentation:** 3
**Contribution:** 2
**Rating:** 4
**Confidence:** 3

**Summary:**

This work presents WS-KAN, a meta-network tailored for working with Kolmogorov-Arnold Networks (KANs). The authors introduce a graph representation (KAN-graph) for KANs that accounts for their permutation symmetries, and they employ a GNN structure built on this encoding. They conduct a theoretical examination of the model's expressive capabilities and perform empirical evaluations using a freshly assembled collection of pre-trained KANs across multiple tasks, reliably showing better results than competing methods.

**Strengths:**

- This paper introduce the first metanetwork for KANs.
- The motivation and design process are presented clearly.
- The KAN model zoo (covering INR classification, accuracy prediction, pruning) provides a valuable benchmark for future WS models on KANs.

**Weaknesses:**

- The discussion in Section 3.1 on permutation symmetries in the KAN weight space primarily pertains to symmetries in the space of neuron functions and their connections rather than the underlying weight parameters that define those functions. This echoes themes from prior work on weight space symmetries in MLPs [1] and only use permutation symmetries. Could other symmetries extend to KANs?
- Clarification is needed on the scope of each KAN-graph: How is node features initialized, is it associated to input data? If yes, this suggests the WS-KAN operates beyond pure weight space, potentially incorporating input-specific adaptations, which somewhat contradict to the claim being a weight-space-only method.
- In Lemma 4.2, please clearly define what is meant by "the nodes in the first layer are enhanced with the input x", does this imply concatenation, embedding, or another operation, and how does it affect the overall model and expressivity?
- The paper focuses on B-spline parameterization for KAN neurons, but it would be beneficial to comment on alternative parameterizations and whether the proposed meta-network generalizes to them.
- The experiments lack reporting on time and memory complexity. Additionally, how well does the approach scale to generating larger models (eg, deeper and wider KANs)
- As the WS-KAN is GNN-based, I wonder if the method can be tested for generalization, eg, by training the meta-network on small KANs and evaluating its ability to generate parameters for larger, unseen KAN architectures?
- Experiments illustrating the importance of positional encoding (if used in the graph representation) would be valuable, perhaps through an ablation study.

Reference:

[1] Zhao et al. Symmetry in Neural Network Parameter Spaces. arxiv:2506.13018.

**Questions:**

See weaknesses.

---

> ### Author Response · Authors · 2025-11-21
> **Response to reviewer 1aub [1/3]**
>
> We thank the reviewer for the constructive feedback. We are pleased that they appreciate our work as the first to address weight space models for KANs and the model zoo we constructed, which can be a valuable benchmark for future work. Below, we address the reviewer's questions and comments.
>
> ---
>
> ***"The discussion in Section 3.1 on permutation symmetries in the KAN weight space primarily pertains to symmetries in the space of neuron functions and their connections rather than the underlying weight parameters that define those functions. This echoes themes from prior work on weight space symmetries in MLPs [1] and only use permutation symmetries. Could other symmetries extend to KANs?"***
>
> We thank the reviewer for bringing this up. We haven't found any other symmetries that extend to KANs. For example, the scaling symmetries that are present in MLPs and CNNs, are not present in the standard KANs, as its 1D functions are typically based on B-splines, which are not invariant to scaling. We believe this is an interesting question for future work -- as KANs continue to evolve, their 1D functions will likely change and introduce more symmetries, which will be interesting to explore.
>
> ---
>
> ***"Clarification is needed on the scope of each KAN-graph: How is node features initialized, is it associated to input data? If yes, this suggests the WS-KAN operates beyond pure weight space, potentially incorporating input-specific adaptations, which somewhat contradict to the claim being a weight-space-only method."***
>
> We apologize for the confusion, and thank the reviewer for the opportunity to clarify. We note that in all experiments we have conducted, the input data is not used to initialize the node features, thus making our architecture a weight space only method.
>
> The only exception to this, is for the theoretical results we present in Proposition 4.2, where we assume that the nodes in the first layer are enhanced with the input $x$. We have to do that in this setup since we need to pass the input to the model somehow. We note that this assumption is consistent with previous work on weight space models for standard architectures, which uses the input to the model in their proof of approximating a forward pass (e.g., see [1,4]).
>
> ---
>
> ***"In Lemma 4.2, please clearly define what is meant by "the nodes in the first layer are enhanced with the input x", does this imply concatenation, embedding, or another operation, and how does it affect the overall model and expressivity?"***
>
> We believe the reviewer is referring to Proposition 4.2, as we do not have a Lemma 4.2 in the paper. In order to prove Proposition 4.2, which states that our architecture, WS-KAN, can simulate a forward pass of the input KAN, we need to assume that the nodes in the first layer are enhanced with the input $x$, simply as additional node features. Importantly, we highlight that this assumption is consistent with previous work on weight space models for standard architectures, which uses the input to the model in their proof of approximating a forward pass (e.g., see [1,4]). Importantly, we do not augment the node features with input information in any of our experiments.
>
> ---
>
> ***"The paper focuses on B-spline parameterization for KAN neurons, but it would be beneficial to comment on alternative parameterizations and whether the proposed meta-network generalizes to them."***
>
> We thank the reviewer for bringing this up. We note that the proposed WS-KAN architecture is general and can be applied to any 1D function parameterization that is based on a linear function or even a polynomial in arbitrary basis functions. As a concrete example, if one uses a Fourier series parameterization for the 1D functions, as done in [5], our WS-KAN architecture can work on it out of the box, by simply encoding the coefficients of the Fourier series as edge features.
>
> ---
>
> ***References:***
>
> [1] Graph Metanetworks for Processing Diverse Neural Architectures. ICLR 2024.
>
> [2] Git Re-Basin: Merging Models modulo Permutation Symmetries. ICLR 2023.
>
> [3] KAN: Kolmogorov-Arnold Networks. ICLR 2025.
>
> [4] Equivariant Architectures for Learning in Deep Weight Spaces. ICML 2023.
>
> [5] Kolmogorov-Arnold Fourier Networks. Arxiv 2025.

---

> > ### Author Response · Authors · 2025-11-21
> > **Response to reviewer 1aub [2/3]**
> >
> > ***"The experiments lack reporting on time and memory complexity. Additionally, how well does the approach scale to generating larger models (eg, deeper and wider KANs)"***
> >
> > As a response to this reviewer's question, we took the following steps:
> >
> > 1. We conducted additional generalization experiments for our proposed WS-KAN architecture, to evaluate its generalization to larger KANs.
> >
> > 2. We computed the running time of all experiments regarding the INR classification task.
> >
> > 3. We provide a complexity analysis of our proposed WS-KAN architecture.
> >
> > Below we provide more details for each of the steps above.
> >
> > 1. To evaluate WS-KAN’s ability to generalize to larger KAN architectures, we conducted additional experiments on the INR classification task using the MNIST and Fashion-MNIST datasets. We trained WS-KAN on KANs with architecture **[2 → 32 → 32 → 10]** and tested its performance on the following larger, previously unseen architectures:
> >
> > - (1) **[2 → 48 → 48 → 10]**,
> > - (2) **[2 → 64 → 64 → 10]**,
> > - (3) **[2 → 80 → 80 → 10]**, and
> > - (4) **[2 → 96 → 96 → 10]**.
> >
> > The test accuracies are reported in the table below.
> >
> > | Dataset        | Acc.  [2 -> 32 -> 32 -> 10]       | Acc. [2 -> 48 -> 48 -> 10]       | Acc. [2 -> 64 -> 64 -> 10]       | Acc. [2 -> 80 -> 80 -> 10]       | Acc. [2 -> 96 -> 96 -> 10]       |
> > |----------------|----------------|-----------------|-----------------|-----------------|-----------------|
> >  | MNIST          | 94.3 ± 0.5  | 91.4 ± 0.5   | 81.0 ± 3.2   | 67.0 ± 4.3   | 57.1 ± 6.1   |
> > | Fashion MNIST  | 84.6 ± 0.6  | 84.6 ± 0.6   | 84.3 ± 0.7   | 83.3 ± 0.8   | 82.2 ± 0.7   |
> >
> > We find that WS-KAN maintains reasonably *strong performance even on these larger, unseen architectures, indicating that it can generalize beyond its training size.* As expected, performance gradually decreases as the network width increases, reflecting the increasing difficulty of generalizing to substantially larger KANs. We thank the reviewer for this insightful suggestion, and we have added it to the edited version of the paper, in blue, to Appendix C.4.
> >
> > 2. We note that the running time we observed in the INR classification tasks is the standard run-time observed when using standard Graph Neural Networks (GNNs) on standard graph benchmarks. We report the running time (in seconds) per epoch for both training and testing in the table below, averaging over 10 epochs with standard deviation.
> >
> > | Dataset                | MNIST          | F-MNIST        | CIFAR-10       |
> >  |------------------------|----------------|----------------|----------------|
> > | Train time per epoch (s) | 10.12 ± 0.06   | 10.18 ± 0.05   | 10.88 ± 0.10   |
> >  | Test time per epoch (s)  | 2.84 ± 0.12    | 2.81 ± 0.06    | 3.05 ± 0.08    |
> >
> > We observe that the run-time is contained -- around 10 seconds per epoch for training and 3 seconds per epoch for testing.  We have added this to the edited version of the paper, in blue, to Appendix C.6.
> >
> > 3. Below we provide a complexity analysis of our proposed WS-KAN architecture.
> >
> > We assume the input KAN contains $n_l$ nodes at layer $l \in \{ 1, \ldots, L \} $, and that all nodes and edges share the same feature dimensionality and it is a constant. The total number of nodes is therefore
> >
> > $$
> > N = \sum_{l=1}^L n_l,
> > $$
> >
> > and the total number of edges -- due to full connectivity between consecutive layers -- is
> >
> > $$
> > E = \sum_{l=1}^{L-1} n_l \, n_{l+1}.
> > $$
> >
> > Referring to Equations (a–d) in the paper (lines 226–229), we obtain the following:
> >
> > - **(a)** The function $\texttt{MLP}^{(1;F)}_v$ is applied once per edge, giving $E$ operations. Aggregating over the incoming neighbors of each node contributes another $E$ operations, since the graph is directed and each edge is counted once. Thus, step (a) costs $O(E)$.
> >
> >  - **(b)** This step mirrors (a), except that the edge directions are transposed. Therefore, it also costs $O(E)$.
> >
> > - **(c)** This step applies an MLP independently to each edge, contributing another $O(E)$ operations.
> >
> > - **(d)** This step is a per-node operation, giving a cost of $O(N)$. Because $E > N$, the edge-wise computations dominate the total cost.
> >
> > Hence, the overall complexity of a single layer of WS-KAN is $$ O(E). $$
> >
> > We have added this analysis to the edited version of the paper, in blue, to Appendix C.6.

---

> ### Author Response · Authors · 2025-11-21
> **Response to reviewer 1aub [3/3]**
>
> ***"As the WS-KAN is GNN-based, I wonder if the method can be tested for generalization, eg, by training the meta-network on small KANs and evaluating its ability to generate parameters for larger, unseen KAN architectures?"***
>
> Please refer to our response to the previous reviewer's question above. In short our new experiments show that our model can generalize very well to unseen architectures.
>
> ---
>
> ***"Experiments illustrating the importance of positional encoding (if used in the graph representation) would be valuable, perhaps through an ablation study."***
>
> We thank the reviewer for bringing this up, and we conducted an ablation study to evaluate the importance of the positional encoding. Specifically, we have rerun all experiments in the paper without the positional encoding, and we report the results in the tables below.
>
> 1. INR Classification Accuracy
>
> | Model              | MNIST         | F-MNIST       | CIFAR-10     |
> |---------------------|---------------|---------------|--------------|
>  | **WS-KAN – no PE**  | **94.3 ± 0.4** | 84.5 ± 0.1 | **46.1 ± 1.2** |
>  | **WS-KAN**          | **94.3 ± 0.5**    | **84.6 ± 0.6**    | 42.2 ± 0.8    |
>
> 2. Accuracy prediction
>
> | Models | MNIST  (MSE)             | F-MNIST (MSE)             | K-MNIST (MSE)             |
> |-----------|----------------------|----------------------|----------------------|
>  | **WS-KAN – no PE**  | **2.53 ± 0.22**    | **2.44 ± 0.50**    | **1.1 ± 0.17**    |
> | **WS-KAN**   | 3.29 ± 0.17          | 2.94 ± 0.13          | 1.45 ± 0.08          |
>
>  | Model | MNIST  (R²)             | F-MNIST (R²)             | K-MNIST (R²)             |
> |-----------|----------------------|----------------------|----------------------|
> | **WS-KAN – no PE**  | **96.00 ± 0.35**    | **93.57 ± 1.32**    | **96.62 ± 0.51**    |
> | **WS-KAN**| 94.81 ± 0.27         | 92.27 ± 0.35         | 95.69 ± 0.24         |
>
> 3. Pruning Mask Prediction
>
> | Model | MNIST  (Accuracy)             | F-MNIST (Accuracy)             | K-MNIST (Accuracy)             |
> |-----------|----------------------|----------------------|----------------------|
> | **WS-KAN – no PE** | 97.77 ± 0.28    | 98.68 ± 0.04    | 96.28 ± 0.45    |
> | **WS-KAN**| **97.93 ± 0.19**         | **98.93 ± 0.05**         | **97.72 ± 0.14**         |
>
>  | Model | MNIST  (AUC)             | F-MNIST (AUC)             | K-MNIST (AUC)             |
> |-----------|----------------------|----------------------|----------------------|
> | **WS-KAN – no PE** | 99.51 ± 0.09    | 99.59 ± 0.02    | 98.85 ± 0.21    |
>  | **WS-KAN**| **99.54 ± 0.01**         | **99.72 ± 0.02**         | **99.46 ± 0.09**       |
>
> We observe that WS-KAN performs well even without positional encoding. However, on the more challenging equivariant task -- pruning mask prediction -- WS-KAN with positional encoding outperforms the version without it in all 6 out of 6 cases. This indicates that positional encoding plays a more important role in enhancing WS-KAN’s performance on equivariant tasks.  We have added this to the edited version of the paper, in blue, to Appendix C.5.

---

### Official Review · Reviewer_gQZG · 2025-10-31

**Soundness:** 3
**Presentation:** 3
**Contribution:** 2
**Rating:** 4
**Confidence:** 4

**Summary:**

This paper introduces applying weight-space symmetric architecture for Kolmogorov–Arnold Networks (KANs), called WS-KAN, which leverages a graph representation of the network’s computation to respect inherent symmetries in the architecture. The authors highlight the equivalence of permutation symmetries in KANs to those in standard neural networks, especially MLPs, and propose the KAN-graph—a directed graph where nodes represent neurons and edges carry univariate functions. WS-KAN uses existing work of graph metanetwork to process these graphs and simulate the forward pass of the original KAN. Empirical validation is conducted using a model zoo of pre-trained KANs across various tasks. The proposed model consistently outperforms structure-agnostic baselines and provides superior performance in INR classification, accuracy prediction, and pruning mask prediction.

**Strengths:**

The method is the first to specifically address the design of weight-space models for KANs.

The authors construct a comprehensive "model zoo" of KANs and benchmark WS-KAN across various tasks, demonstrating superior performance over baselines. The extensive experiments on multiple datasets such as MNIST, CIFAR10, and F-MNIST validate the proposed method's robustness.

**Weaknesses:**

The paper basically uses the existing solution of graph metanetwork proposed for MLPs to apply on KAN weight space, which is intuitive, however contribution is limited. Also experiments e.g. INR classification, accuracy prediction, all follow the standard from weight space symmetry works on MLP/CNN. From the technical solution and application tasks I do not see anything new and specific to KANs.

**Questions:**

- What does proposition 3.2 imply and how it is related to the main contributions / arguments in the paper? Here "the nodes in the first layer are enhanced with the input x", do you really do that in the implementation?

- Have you tried other weight space symmetry works such as DWSNet and NFN, and do they apply to KANs as well?

---

> ### Author Response · Authors · 2025-11-21
> **Response to reviewer gQZG [1/2]**
>
> We thank the reviewer for the constructive feedback.
> We are pleased that they appreciate our work as the first to address weight space models for KANs, the model zoo we constructed, and the comprehensive experiments we conducted.
> Below, we address the reviewer's questions and comments.
>
> ---
>
> ***"The paper basically uses the existing solution of graph metanetwork proposed for MLPs to apply on KAN weight space, which is intuitive, however contribution is limited"***
>
> We appreciate the reviewer’s comment and recognize that our contributions were not clearly communicated.
>
> Although graph metanetworks exist for MLPs, they do not directly transfer to KANs because KAN weights are parametrized via 1D functions rather than scalars. This makes encoding the full functional information impractical. The central challenge, therefore, is to design a graph construction that respects the KAN structure, while remaining expressive and performing well empirically when processed with WS-KAN.
>
> To address this challenge, we use only the coefficients composing the 1D functions to define the weights on the edges. Our key contribution is showing that this simple and elegant construction – despite omitting full functional information – is both theoretically sufficient and empirically effective. We prove that this compact representation can approximate the underlying 1D functions (Lemma 4.1 in the paper) and can simulate the KAN forward pass (Proposition 4.2 in the paper), preserving the expressive power of standard graph metanetworks [1,4]. We further validate this construction with a comprehensive set of experiments on INR classification, accuracy prediction, and a KAN-specific pruning-mask prediction task.
>
> Importantly, we wish to also point the reviewers' attention to an additional contribution in Appendix D -- Aligning Kolmogorov-Arnold Networks – in which we introduce an extension to the alignment algorithm of standard MLP architectures [2], so it can now also be applied to KANs. We believe this extension is another (minor) contribution of our paper, as we also experimented with this alignment algorithm and found it to be a strong baseline for weight space models for KANs (outperforming MLP and MLP+Augmentation on most tasks).
>
> We will make sure to convey our contributions better in the revised version.
>
> ---
>
> ***"Also experiments e.g. INR classification, accuracy prediction, all follow the standard from weight space symmetry works on MLP/CNN. From the technical solution and application tasks I do not see anything new and specific to KANs."***
>
> As previously acknowledged by the reviewer, we are the first to design, analyze, and test a weight space model for KANs. To provide a thorough evaluation of our proposed method, we chose to use both standard tasks from the weight space learning literature for MLPs and CNNs—INR classification and accuracy prediction—for which we constructed a model zoo, and a *novel task of pruning mask prediction, which is tailored specifically to KANs*, which we believe represents a new contribution to the field.
>
> Let us explain the pruning mask prediction task in more detail. We start by training a dataset of KANs and their corresponding pruning masks, using the original KAN pruning technique as outlined in [3] to obtain this masking. This pruning algorithm is data-driven, namely, it prunes the KANs activations (e.g., edges) based on the activation values of the 1D functions based on the dataset. We collect this dataset of KANs and their pruned counterparts to serve as our supervision.
>
> We train our WS-KAN model to predict the proper masking for new KANs. As shown in Figure 5 in our paper, using our WS-KAN for pruning nearly matches the performance of the original pruning technique, making it a viable approach in practice. Moreover, our method is 5 orders of magnitude faster than the standard pruning technique (see Figure 5c in our paper), making it significantly more practical for real-world applications.
>
> We believe this thorough analysis of WS-KAN, which also draws inspiration from the standard tasks of weight space symmetry for MLPs and CNNs, and the additional novel task of pruning mask prediction which is tailored specifically to KANs, represents a new contribution to the fields of both weight space models, and KANs specifically.
>
> ---
> ***References:***
>
> [1] Graph Metanetworks for Processing Diverse Neural Architectures. ICLR 2024.
>
> [2] Git Re-Basin: Merging Models modulo Permutation Symmetries. ICLR 2023.
>
> [3] KAN: Kolmogorov-Arnold Networks. ICLR 2025.
>
> [4] Equivariant Architectures for Learning in Deep Weight Spaces. ICML 2023.
>
> [5] How Powerful are Graph Neural Networks?. ICLR 2019
>
> [6] Weisfeiler and Leman Go Neural: Higher-order Graph Neural Networks. AAAI 2019.

---

> > ### Author Response · Authors · 2025-11-21
> > **Response to reviewer gQZG [2/2]**
> >
> > ***"What does proposition 3.2 imply and how it is related to the main contributions / arguments in the paper?***
> >
> > We believe the reviewer is referring to Proposition 4.2, as the paper contains no Proposition 3.2. Proposition 4.2, which shows that our design choice of WS-KAN can simulate a forward pass of the input KAN, is essential, as we elaborate below.
> >
> > We note that equivariant architectures in general  – such as WS-KAN – have more limited expressive power compared to non-equivariant architectures such as MLPs, this is by design as we would like to work on a reduced hypothesis space for better generalization. In many cases, and particularly for GNNs [5,6], this restriction overconstrains the space of functions, making it effectively too small, and not being able to approximate important functions.
> >
> > Because our proposed equivariant WS-KAN does not encode the full functional information of an input KAN (as this is impractical) it is even more important to understand its expressivity. *Proposition 4.2 addresses precisely this point: it shows that WS-KAN preserves the expressive guarantees established for weight-space models in standard architectures (e.g., [1, 4]).* This result also lays the groundwork for future, more general expressivity analyses. For example, being able to approximate the forward pass means that WS-KAN can separate input KANs that represent different functions. This, in turn, opens the door to using strong theorems, such as the Stone-Weierstrass theorem that builds on such separation results.
> >
> > ---
> >
> > ***"'the nodes in the first layer are enhanced with the input x' do you really do that in the implementation?"***:
> >
> > We note that in practice, we do not enhance the nodes in the first layer with the input $x$. However, in the settings of Proposition 4.2 we need to compute a forward pass of the input KAN, and hence we need to pass  the input $x$ to the KAN somehow. Therefore, for the sake of the proof of Proposition 4.2, we assume that the nodes in the first layer are enhanced with the input $x$. We highlight that this assumption is consistent with previous work on weight space models for standard architectures, which uses the input to the model in their proof of approximating a forward pass (e.g., see [1,4]).
> >
> > ---
> >
> > ***"Have you tried other weight space symmetry works such as DWSNet and NFN, and do they apply to KANs as well?"***
> >
> > We note that applying other weight space techniques to KANs is not straightforward and requires careful consideration. This stems from the fundamental difference that, for MLPs, the weights are matrices of scalars, whereas for KANs, we have 1D functions rather than scalar values. Applying DWSNet or NFN techniques to KANs would require proper adaptation to handle functions instead of scalars. Such an adaptation would fundamentally change both the architecture, the performance, and the theoretical analysis, potentially resulting in different expressive power compared to DWSNet or NFN applied to MLPs.
> >
> > ---
> > ***References:***
> >
> > [1] Graph Metanetworks for Processing Diverse Neural Architectures. ICLR 2024.
> >
> > [2] Git Re-Basin: Merging Models modulo Permutation Symmetries. ICLR 2023.
> >
> > [3] KAN: Kolmogorov-Arnold Networks. ICLR 2025.
> >
> > [4] Equivariant Architectures for Learning in Deep Weight Spaces. ICML 2023.
> >
> > [5] How Powerful are Graph Neural Networks?. ICLR 2019
> >
> > [6] Weisfeiler and Leman Go Neural: Higher-order Graph Neural Networks. AAAI 2019.

---

### Author Response · Authors · 2025-11-21
**General response**

We are grateful to all reviewers for their feedback and constructive comments. We are particularly encouraged that reviewers recognized our work as the first to investigate weight-space models for KANs:

- *"The method is the first to specifically address the design of weight-space models for KANs."* (**gQZG**)
- *"This paper introduces the first metanetwork for KANs."* (**1aub**)

The reviewers also noted our method’s *“robustness”* (**gQZG**) and its *“superior performance over baselines”* (**gQZG**). They have also found our method to be *“explained and motivated”* (**y2se**, **1aub**), noting: *“The paper is well-written and self-contained… I was able to understand the architecture clearly… clear explanation of how a KAN can be interpreted as a graph”* (**vi4a**). Finally, they have also appreciated our symmetry analysis *“The authors provide a good demonstration of the symmetry properties of the KAN parameter space”* (**vi4a**) and the theoretical results we have obtained (**y2se**).

We also appreciate the reviewers’ positive reception of the KAN model zoo we constructed for benchmarking weight-space models:

- *"The authors construct a comprehensive "model zoo" of KANs ..."* (**gQZG**)
- *"The KAN model zoo (covering INR classification, accuracy prediction, pruning) provides a valuable benchmark for future WS models on KANs."* (**1aub**)

---

Taking into consideration all the comments, suggestions, and other feedback provided by the reviewers, we have conducted additional experiments that we believe have significantly strengthened our paper, which we added to the final version of our paper in blue:

1. As noted by (**1aub**), we evaluated the ability of our WS-KAN architecture to generalize to larger KANs that were not seen during training, and **we found that the method generalizes remarkably well to these larger models**. *This part is added in the edited version of the paper, in Appendix C.4, in Table 5.*

2. In addition, following the requests of (**y2se**) and (**1aub**), we conducted an ablation study examining the contributions of our positional encoding and bidirectional message passing. *The corresponding tables are included in this rebuttal and are added in the edited version of the paper, in Appendix C.5, Tables 5,6,7,8,9,10.*

3. Finally, as requested by (**y2se**), we provide a computational complexity analysis together with runtime measurements, showing that WS-KAN achieves runtime performance comparable to standard Graph Neural Networks (GNNs) on widely used graph benchmarks. *This is also added to the edited version of the paper, in Appendix C.6, Table 11.*

---

### Author Response · Authors · 2025-12-03
**Summary message for re-assigned AC  [1/2]**

We are grateful to the Senior AC, the initially assigned AC, and all reviewers for their contributions so far. Their feedback highlighted important points that ultimately strengthened our work. Through the rebuttal process, **we addressed all reviewer questions and ran the additional experiments they requested. These efforts have strengthened the paper, and resulted in increased scores.**

---

**In this first thread, we provide a short summary of the discussion period until it was interrupted. The next thread offers additional details, should the AC have time.**

At the beginning of the rebuttal, our scores were 8 (vi4a), 4 (y2se), 4 (1aub), 4 (gQZG).
After our detailed responses, the reviewers replied as follows:

- **vi4a acknowledged maintaining the score of 8.**
- **y2se raised his score to 6.**
- **The other two reviewers had not responded to our rebuttal before the discussion period was interrupted.**

**This brings our paper's average to 5.5, with scores:**

- **8 (vi4a)** - **participated** in the discussion period, and acknowledged maintaining his 8.
- **6 (y2se)** - **participated** in the discussion period and raised his score.
- **4 (1aub)** - **did not participate** in the discussion before it was intrrupted.
- **4 (gQZG)** - **did not participate** in the discussion before it was intrrupted.

Beyond this point, the reviewers identified a number of notable strengths in our paper, in particular, **acknowledging that we are the first to design weight-space models for KANs, and appreciating our comprehensive model zoo and strong empirical results**:

 - *"The method is the first to specifically address the design of weight-space models for KANs."*  (gQZG)
- *"This paper introduces the first metanetwork for KANs."* (1aub)
 - *"The authors construct a comprehensive "model zoo" of KANs ..."* (gQZG)
- *"The KAN model zoo (covering INR classification, accuracy prediction, pruning) provides a valuable benchmark for future WS models on KANs."* (1aub)
- *“The authors provide a good demonstration of the symmetry properties of the KAN parameter space”* (vi4a)
- The *"robustness"* of our method (gQZG)
- Our *“superior performance over baselines”* (gQZG)
- They found our method to be *“explained and motivated”* (y2se, 1aub)
- *“The paper is well-written and self-contained… I was able to understand the architecture clearly… clear explanation of how a KAN can be interpreted as a graph”* (vi4a).
- They also appreciated the theoretical results we have obtained (y2se).


**During the rebuttal phase, we carried out additional experiments and addressed all requests from the reviewers.** Should the AC’s time availability permit, we provide below further details to support the final decision, including a summary of the reviewers’ comments and questions along with our responses.

---

> ### Author Response · Authors · 2025-12-03
> **Summary message for re-assigned AC [2/2]**
>
> Below we present a summary of the reviewers comments and questions, along with a summary of our responses.
>
> ---
>
> **Generalization to larger KANs.**
>
> To address Reviewer 1aub’s request, **we ran additional experiments on two datasets** testing how well our architecture scales to KANs larger than those seen during training. As shown in our revised version Appendix C.4, Table 5 in blue (and in the rebuttal), **the results were positive: our method generalizes effectively to much larger input KANs**.
>
> ---
>
> **Ablation studies.**
>
> In response to reviewers y2se and 1aub, **we repeated all experiments in the paper ablating over the positional encoding and the bidirectional message-passing components.** The results (available in Appendix C.5, Tables 6-10 in the revised version in blue, and in the rebuttal), showcase that: **(1) bidirectional message passing is consistently essential, and (2) positional encoding matters more for harder (equivariant) tasks, while performance remains strong without it on simpler tasks**.
>
> ---
>
> **Complexity and run-time analysis.**
>
> In response to reviewers y2se and 1aub, we added a complexity and run-time analysis, both in the rebuttal, and in Appendix C.6, Table 11 (in the revised version in blue). **The overall complexity is $O(E)$**, where $E$ is the number of edges of the input KAN, with **run-times of about $10$ seconds per training epoch and $3$ seconds per testing epoch – typical** for message-passing models on standard graph benchmarks.
>
> ---
>
> **Empirical validation of the invariance/equivariance of our model w.r.t. the symmetry.**
>
> Reviewer y2se requested empirical validation of the equivariance of our model. **We provided a stronger validation** - in the beginning of "Response to reviewer y2se [1/3]" **we (sketch) prove that:** *any permutation invariant GNN architecture (of which our model is a special case of) is guaranteed to produce exactly the same output for any two input KANs that differ only by a permutation of their intermediate nodes.*
>
> ---
>
> **Empirical evaluation attributed to KANs.**
>
> Reviewers y2se and gQZG asked for clearer distinctions between our setup and prior weight-space models for MLPs. We emphasized that, as they noted, **ours is the first weight-space model designed and evaluated for KANs.** To thoroughly assess it, **we used both standard weight-space tasks (which are typically used to assess weight-space models for standard neural networks)** – INR classification and accuracy prediction, **of which we created a model zoo for from scratch – and a new KAN-specific task: edge-pruning mask prediction**, which we believe is **a meaningful additional contribution with practical relevance for real-world KAN pruning.**
>
> ---
>
> **Clarification regarding Proposition 4.2, and its contribution to the paper.**
>
> Reviewer gQZG questioned the contribution of Proposition 4.2 -- which shows that our WS-KAN can approximate a forward pass given an input KAN and its input. We clarified that equivariant architectures (such as our WS-KAN) intentionally trade expressive power for improved generalization by restricting the hypothesis space, which can sometimes over-constrain models, making expressivity analysis essential. **Since a weight space model for KANs cannot feasibly encode the full functional information of a KAN (unlike weight-space models for standard neural networks), Proposition 4.2 plays a key role: it shows that our design of WS-KAN preserves the fundamental expressive guarantees established for prior weight-space approaches.** It also sets the stage for future extensions, as the ability to approximate the forward pass ensures WS-KAN can distinguish KANs representing different functions, enabling stronger theoretical results such as those derived from Stone–Weierstrass.
>
> ---
>
> **Additional comments.**
>
> Reviewers asked about exploring alternative weight-space techniques (gQZG, vi4a), looking into other potential symmetries in KANs (1aub), and considering KANs with different 1D parameterizations beyond B-splines (1aub). We view all of these as interesting directions for future work.

---

### Meta-Review · Area_Chair_F2it · 2025-12-28

**Summary:**

gQZG: (1) contribution is limited. (2) nothing new/specific to KANs

1aub: (1) discussion in Sec 3.1 on symmetries is limited. (2) need clarification on the scope of each KAN-graph. (3) Lemma 4.2 needs clarification. (4) Need discussion on alternative parameterizations and whether the method generalizes to them. (5) Need to report time/memory complexity in experiments. Also need experiments on larger models. (6) Need experiments on generalization for larger, unseen KAN architectures. (7) Need ablation study on positional encoding.

y2se: (1) Lack of empirical symmetry validation. (2) Limited range of equivariant tasks. (3) scalability issue. (4) need ablation study to justify design choices.

vi4a: question regarding other types of symmetry

**Reviewer Concerns:**

gQZG: this reviewer's concerns are addressed (although the reviewer has not explicitly responded)

1aub: the concerns are addressed (although the reviewer has not explicitly responded)

y2se: the concerns are addressed and the reviewer responded and agreed to increase the score 4->6.

vi4a: this reviewer is mostly positive and maintains his/her score.

**Reviewer Scores:**

One reviewer (y2se) explicitly mentioned increasing the score 4->6. Another reviewer (vi4a) maintained his/her original positive score 8.

Two reviewers (gQZG, 1aub) have not respond yet. But based on the rebuttal, it is likely their scores will increase.

---

### Decision · Program_Chairs · 2026-01-26

Accept (Poster)